# Nitrite regeneration in the oligotrophic Atlantic Ocean

Darren R. Clark[1], Andrew P. Rees[1], Charissa M. Ferrera[2], Lisa Al-Moosawi[1], Paul J. Somerfield[1], Carolyn Harris[1], Graham D. Quartly[1], Stephen Goult[1], Glen Tarran[1], Gennadi Lessin[1].

[1]Plymouth Marine Laboratory, Prospect Place, The Hoe, PL1 3DH, UK.

[2]The Marine Science Institute, Velasquez St., University of the Philippines, Diliman, Quezon City 1101 Philippines.

*Correspondence to:* Andrew Rees (apre@pml.ac.uk)

**Abstract.** The recycling of scarce nutrient resources in the sunlit open ocean is crucial to ecosystem function. Nitrification directs ammonium ($NH_4^+$) derived from organic matter decomposition towards the regeneration of nitrate ($NO_3^-$), an important resource for photosynthetic primary producers. However, the technical challenge of making nitrification rate measurements in oligotrophic conditions combined with the remote nature of these environments means that data availability, and the understanding that provides, is limited. This study reports nitrite ($NO_2^-$) regeneration rate ($R_{NO2}$ - the first product of nitrification derived from $NH_4^+$ oxidation) over a 13, 000 km transect within the photic zone of the Atlantic Ocean. These measurements, at relatively high resolution (order 300 km), permit the examination of interactions between $R_{NO2}$ and environmental conditions that may warrant explicit development in model descriptions. At all locations we report measurable $R_{NO2}$ with significant variability between and within Atlantic provinces. Statistical analysis indicated significant correlative structure between $R_{NO2}$ and ecosystem variables, explaining ~65% of the data variability. Differences between sampling depths were of the same magnitude or greater than horizontally resolved differences, identifying distinct biogeochemical niches between depth horizons. The best overall match between $R_{NO2}$ and environmental variables combined chlorophyll-a concentration, light phase duration and silicate concentration (representing a short-term tracer of water column physical instability). On this basis we hypothesise that $R_{NO2}$ is related to the short-term autotrophic production and heterotrophic decomposition of dissolved organic nitrogen (DON), which regenerates $NH_4^+$ and supports $NH_4^+$ oxidation. However, this did not explain the observation that $R_{NO2}$ in the deep euphotic zone was significantly greater in the southern compared to the northern hemisphere. We present the complimentary hypothesis that observations reflect the difference in DON concentration supplied by lateral transport into the gyre interior from the Atlantic's eastern boundary upwelling ecosystems.

## 1. Introduction

Oligotrophic gyres and their transition regions are the largest biome on Earth, covering approximately 60% of its surface (Eppley and Peterson, 1979), and are expanding as a result of the warming climate (Polovina et al., 2008). Biogeochemical processes in these regions have the potential to influence global elemental cycles. Within the sunlit ocean, photosynthetic cells fix carbon dioxide ($CO_2$) and assimilate inorganic nutrients to synthesise organic molecules and build biomass (Azam et al., 1983). Either directly or indirectly, this biomass supports the ecosystem and contributes over half of global organic carbon export (Emerson et al., 2001). Persistent stratification in these systems impedes nutrient inputs to the sunlit region from the deep ocean, constraining microbial growth (Eppley and Peterson, 1979, Moore et al., 2013). Consequently, efficient organic matter decomposition and inorganic nutrient regeneration are crucial to ecosystem function. Decomposition regenerates inorganic nitrogen in the form of ammonium ($NH_4^+$), which rarely accumulates in the open surface-ocean as it is readily used in anabolic and chemotrophic microbial processes. Evidence demonstrates that nitrification, the 2-stage oxidation of $NH_4^+$ via nitrite ($NO_2^-$) to nitrate ($NO_3^-$), takes place in the photic zone of open oceans (Santoro et al., 2013) where it directly competes with photosynthetic cells for the $NH_4^+$ resource (Smith et al., 2014). The outcome of this competitive interaction is important; by moderating the composition and concentration of the dissolved inorganic nitrogen pool ($NH_4^+$, $NO_2^-$, $NO_3^-$), nitrification influences nutrient limitation patterns for primary production at the basin scale (Fawcett et al., 2015; Moore et al., 2013; Palinska et al., 2002; Partensky et al., 1999). Biological processes within the ocean's upper mixed layer modify its chemistry and subsequently influence the composition of deep water masses through particle sedimentation, decomposition and remineralisation, with global implications for productivity and climate over a range of temporal scales (Rafter et al., 2013; Tuerena et al., 2015). In addition, nitrification is a source of the climatically active gas $N_2O$ within the surface ocean, which readily exchanges with the lower atmosphere. Characterising the distribution and constraints on

nitrification is an important step towards fully assessing pelagic nitrogen transformations in the open ocean, although observations at the oceanic basin scale are limited (Clark et al., 2008; Shiozaki et al., 2016).

Within the open oceans, pelagic nitrification is distributed in a characteristic depth profile; low rates are measured in the well-lit near-surface waters while the highest rates are reported at, or close to, the base of the photic zone in association with the primary nitrite maximum (PNM) and the deep chlorophyll maximum (DCM); rates drop rapidly below this depth into the twilight zone of the mesopelagic (Newell et al., 2013; Smith et al., 2016). Observational data have been crucial to the development of this understanding and indicate that environmental factors act simultaneously to structure the vertical distribution of nitrifying organisms and their activity. Evidence identifies roles for light (Olson, 1981; Shiozaki et al., 2016), $NH_4^+$ concentration, (Smith et al., 2016), organic matter flux (Benner and Amon, 2015; Newell et al., 2011) and competitive interactions with phytoplankton (Fawcett et al., 2015; Smith et al., 2014). However, the statistical analysis of nitrifying organism activity and diversity in relation to environmental factors explains only a fraction of the data variability, alluding to further complex and undefined interactions between the nitrifying activity of these organisms and their environment (Bouskill et al., 2012; Clark et al., 2014). A better appreciation of the factors that most significantly influence nitrifying activity is required to guide future fieldwork observations.

Here we present relatively high-resolution photic zone measurements of $NO_2^-$ regeneration rate ($R_{NO2}$, the product of $NH_4^+$ oxidation, the first stage of nitrification) across a 13,000 km oligotrophic Atlantic Ocean transect to examine the process rate distribution and potential links to environmental conditions. Investigations were conducted during the Atlantic Meridional Transect program (Rees et al., 2017), which, since 1995, has undertaken annual (or bi-annual) scientific observations of the Atlantic Ocean between the UK and southern hemisphere locations including the Falkland Islands and Punta Arenas (Chile). The cruise track has historically navigated through marine provinces that include the Atlantic gyres, the equatorial upwelling convergence zone, mesotrophic fringing regions and regions influenced by eastern boundary upwelling ecosystems (Mauritania, Benguela), sampling significant variability in biological, physical and chemical conditions. By navigating between hemispheres, observations incorporate contrasting seasons, a signature that is evident in biogeochemical observations, despite persistent water-column stability that typifies tropical and sub-tropical open oceans. Within this observational context, the overarching objective of the study was to highlight interactions between $R_{NO2}$ and environmental conditions that may warrant explicit development and inclusion in model descriptions.

## 2. Methods

### 2.1 Water column characterization.

This study was undertaken during the Atlantic Meridional Transect (AMT) cruise 19 (13[th] October to 1[st] December 2009) aboard the RRS James Cook between Falmouth (UK) and Punta Arenas (Chile; Fig. 1), passing through the following Atlantic provinces (Longhurst, 1998); North Atlantic Drift (NADR), North Atlantic subtropical gyre (NAST), North Atlantic tropical gyre (NATR), Western tropical Atlantic (WTRA), South Atlantic gyre (SATL). Seawater was collected for biogeochemical observations from a total of 120 CTD-rosette casts, of which 33 were used for isotope studies to derive $R_{NO2}$. An instrumented package was deployed during solar noon to derive semi-continuous vertical profiles of light, salinity, temperature, chlorophyll concentration and dissolved oxygen. Light depth profiles were derived using Photosynthetically Active Radiation (PAR) data recorded by a Chelsea Instruments PAR sensor. A stainless-steel rosette unit was deployed during dawn casts (typically 06:00 GMT), incorporating a Sea-Bird 9/11 plus Conductivity, Temperature, Depth system, a Chelsea MKIII Aquatracka fluorimeter, a Sea-Bird SBE 43 Dissolved Oxygen Sensor and 20 L Niskin bottles for water collection from specific sampling depths.

### 2.2 Nutrient analysis.

Inorganic nutrient samples collected from dawn casts at between 6-12 depths within the upper 250 meters were immediately analysed. Nitrite ($NO_2^-$), nitrate ($NO_3^-$), phosphate ($PO_4^{3-}$) and silicate ($Si(OH)_4$) were measured colorimetrically with a 5-channel segmented flow Bran and Luebbe AAIII autoanalyzer, using methods described previously (Woodward and Rees, 2001). The concentration of $NH_4^+$ was not determined during this study.

### 2.3 Analytical Flow Cytometry.

Microbial cells were enumerated based on their light scattering and auto-fluorescence properties using a BD FACSort™ flow cytometer. Photosynthetic cells were analysed live within 1-2 hours of collection. Non-photosynthetic cells were fixed with paraformaldehyde for 1 hour at 4 °C and then stained with SYBR Green I DNA stain (Life Technologies, Paisley, UK) in the dark at room temperature for 1 hour. Live samples were analysed at a flow rate of 180 µL min$^{-1}$ for 4 minutes. Fixed/stained samples were analysed at either 50 or 12 µL min$^{-1}$ for 1-2 minutes, depending on concentration. Data acquisition for photosynthetic cells was triggered on red (chlorophyll) fluorescence while the enumeration of SYBR Green I-stained bacteria and archaea was triggered on green fluorescence. Flow rates were calibrated daily using fluorospheres of known concentration.

### 2.4 $R_{NO2}$ determination using $^{15}$N isotope studies.

Approximately 4 L of seawater collected from depths equivalent to 14 %, 1 % and 0.1 % (the latter defining the base of the euphotic zone for the purposes of this investigation) of surface measured PAR (sPAR) was used for $R_{NO2}$ studies. The isotope dilution approach (Clark et al., 2007; 2008; 2014) was used to derive $R_{NO2}$, by measuring the rate of product formation ($NO_2^-$ regeneration). These 4 L sample volumes were amended with $^{15}NO_2^-$ at < 10 % of the ambient concentration. Following amendment, triplicate 500 mL sub-samples were taken. Within these triplicate sub-samples, pre-incubation $NO_2$ concentration and $^{15}$N enrichment was determined by synthesising Sudan-1 (see below) from sample $NO_2^-$. From the remaining volume, triplicate 500 mL incubation vessels were filled and placed in deck incubation units covered with neutral density filters to simulate sPAR depths according to Joint Global Ocean Flux Study protocols (IOC, 1994). Incubation boxes were flushed with recirculating depth-appropriate temperature-controlled seawater. Following approximately 10 hours of day-time incubation, experiments were terminated by GF/F filtration. Sudan-1 was synthesised from $NO_2^-$ within 500 mL samples, allowing post-incubation $NO_2^-$ concentration and $^{15}$N enrichment to be measured. During these studies, a pre-incubation $^{15}NO_2^-$ enrichment of 5.41±2.18 % was achieved, which was diluted by 0.92±0.83 % on average due to biological activity during incubation.

Sudan-1 is synthesised for this purpose of measuring $NO_2^-$ concentration and isotopic enrichment as it is readily formed and collected at room temperature, it is stable and amenable to both purification and isotopic analysis. For Sudan-1 synthesis, the first reagent (0.8 g of aniline sulphate in 200 mL 3 M HCl) was added to seawater samples in the proportion 0.5 mL per 100 mL sample volume. After mixing, samples were left for 5 minutes to homogenize, after which sample pH was verified to be approximately 2.0. Reagent 2 (24 g NaOH and 0.416 g 2-napthol in 200 mL Milli Q) was added in the proportion 0.5 mL per 100 mL sample volume. Samples were again mixed and left for 5 minutes before sample pH was verified to be approximately 8.0. The development of Sudan-1 was complete after 30 minutes of incubation at room temperature. Samples were acidified with citric acid (final pH 5.5) and an appropriate volume of deuterated Sudan-1 internal standard (100 ng·µL$^{-1}$) was added. Deuterated Sudan-1, which was used for sample quantification, was synthesized and purified according to methods described previously (Clark et al., 2006; 2007). Samples were collected by solid phase extraction (SPE) using octadecyl (C18) 6 mL, 500 mg cartridges. SPE columns were stored at -20 °C until analysis.

At the land-based laboratory, samples eluted from SPE cartridges were purified by high performance liquid chromatography (HPLC). Using a ramping methanol/Milli-Q water profile in combination with a Gemini-NX 5u C18 110A 250×4.6 mm column, sample peaks were resolved. The peak containing Sudan-1 internal standard was retained using a fraction collector while the remaining sample volume was discarded. Fraction-collected samples were dried and stored over anhydrous silica gel until further analysis. Purified silylated samples were analysed by Gas Chromatography Mass Spectrometry (GCMS) using a HP-5ms column (30 m x 0.25 mm internal diameter), with the ramping profiles and extracted ions described in Clark et al. (2007). An analysis of recovery efficiency from SPE cartridges within a theoretical mass range of 200-1000 ng was >94 % on average, decreasing with increased sample size from 99 to 92 %. Within this sample mass range, the standard deviation for the ratio of Sudan-1 ions used for isotopic analysis of $^{15}$N tracer (m/z 306.2/305.2) decreased from 7 ‰ to 1 ‰. Assuming a 200 ng sample of Sudan-1 (although samples as low as 50 ng can be analysed) derived from a 500 mL seawater sample, the associated ambient $NO_2^-$ concentration would be 1.6 nmol L$^{-1}$, significantly less than the average ambient concentration of approximately 5 nmol L$^{-1}$ (see below).

Nitrite regeneration rates ($R_{NO2}$; pmol L$^{-1}$ h$^{-1}$) were calculated according to the Blackburn–Caperon model (Blackburn, 1979; Caperon et al., 1979):

$$R_{NO2} = \left( \frac{\ln \left( \frac{Rt_a}{Ro_a} \right)}{In \left( \frac{St_a}{So_a} \right)} \right) \cdot \left( \frac{(So_a/St_a)}{t_a} \right) \cdot 10^3$$

where $Ro_a$ was the pre-incubation ratio of m/z 306.2/305.2 for sudan-1 derived from sample $NO_2^-$, $Rt_a$ was the post-incubation ratio of m/z 306.2/305.2 for sudan-1 derived from sample $NO_2^-$, $So_a$ was the pre-incubation $NO_2^-$ concentration (nmol N $L^{-1}$) and $St_a$ was the post-incubation $NO_2^-$ concentration (nmol N $L^{-1}$) after $t_a$ hours of incubation.

**2.5 Statistical analysis.**

$R_{NO2}$ data and associated explanatory variables were analysed using a range of multivariate statistical methods (Clarke et al., 2014). Most variables were skewed, so prior to analysis all variables except phosphate excess (P*; calculated as P*=$[PO_4^{3-}]$-($[NO_3^-]$/16)) were log-transformed to reduce the influence of extreme values in subsequent analyses. Variables were normalized by subtracting the mean and dividing by the standard deviation to convert them all to a common dimensionless scale. Relationships among samples based on the explanatory variables were visualised using Principal Components Analysis (PCA), and hypothesis testing for differences among groups of samples was undertaken using Analysis of Similarities (ANOSIM) with ordered factors (Somerfield et al., 2021). The generalised ANOSIM statistic is a scaled measure of effect size comparable across different tests.

To analyse relationships between $R_{NO2}$ and explanatory variables the data for each sPAR band were analysed separately. To determine whether there was structure among variables that warranted further analysis, explanatory variables were subject to Type 2 Similarities Profiles (SIMPROF) analysis (Somerfield and Clarke, 2013). A Euclidean distance matrix was calculated for $R_{NO2}$. A stepwise search algorithm (BVSTEP) with forward selection and backward elimination steps (Clarke and Warwick, 1998) was used to search for the smallest subset of explanatory variables that, in combination, 'best matched' the pattern of $R_{NO2}$ across samples, the best match being defined as maximising the rank correlation between corresponding distance matrices. The algorithm was run with 20 restarts of 6 randomly selected variables. The significance of the maximum rank correlation was tested using an appropriate permutation test (Clarke et al., 2008).

**2.6 Contemporaneous physical oceanography data.**

The ship, RRS James Cook made routine physical measurements whilst underway using the thermosalinograph to give temperature and salinity of the water at ~5m depth, and a depth profile of currents relative to the ship using the vessel-mounted 150 kHz Acoustic Doppler Current Profiler (ADCP). The latter were corrected using the ship's navigation to give absolute instantaneous currents along the ship's track. To put these measurements in context, data were subsequently obtained from satellite altimeter and profiling floats that were in the area. These two sources of data were particularly useful in understanding anomalous observations at two of the CTD stations (28 °S, 39 °S).

The altimetry data were sourced from Copernicus Marine Environment Monitoring Service (CMEMS) who combine the sea surface height measurements from available observations and interpolated these data on to a regular 0.25° x 0.25° grid. Variations in sea level at the mesoscale (order 100-300 km) could be interpreted as eddies, with a local negative anomaly having cyclonic flow about it (clockwise in the southern hemisphere) and highs (anticyclones) having flow in the opposite direction. These flows were in geostrophic balance and thus represent stable current regimes, not incorporating tides or the effect of short wind bursts. Because of the long and near global altimetric record these data can be used to identify regions of persistent variability and to track the evolution of features over periods ranging from weeks to years. However, satellites only give a near-surface view.

Profiles of temperature and salinity in the vicinity of the ship's track were provided by Argo floats, which are autonomous profilers typically recording 0-2000 m vertical sections every 10 days. As these instruments are freely drifting within the ocean, the location and timing of their measurements are not linked to that of the ship. Argo data for the required regions were obtained, and the profiles analysed to give a measure of depth of the surface layer, define as the depth at which temperature was more than 3 °C below that of the surface.

**3. Results**

### 3.1 Temperature, salinity and PAR.

Characteristic physical features of oligotrophic oceans were demonstrated (Fig. 2a-c). The water column was thermally stratified, with sampling locations straddling a temperature range of approximately 12-29 °C. The deepest mixed layer depths were associated with the central regions of the Atlantic gyres. Stratification weakened towards the northern and southern limits of the transect. In terms of salinity, sampling depths incorporated a range of 34.3-37.5. The most saline regions were associated with the ocean gyres and were highest in the Northern gyre. A region of relatively low salinity water was associated with the surface 30 m of the WTRA, defining the lower limit of the salinity range measured. In terms of the light field, 0.1% sPAR defined the deepest sampling point and equated to a depth within the range 75-200 m. Light dose (defined as the light intensity multiplied by the day-light duration) across sampling depths spanned ~2 orders of magnitude. Day light duration increased progressively south, associated with the seasonal transition from boreal autumn to austral spring.

### 3.2 Inorganic nutrients and dissolved oxygen.

Inorganic nutrient concentration measurements demonstrated oligotrophic (< 100 nmol $L^{-1}$ $NO_3^-$ and $PO_4^{3-}$, Fig. 3a and b) conditions throughout the transect at 14 % sPAR. At the 1 % sPAR depth, the concentration of $NO_3^-$ and $PO_4^{3-}$ was generally low (<600 nmol $L^{-1}$ and <150 nmol $L^{-1}$ respectively). The highest concentrations of $NO_3^-$ and $PO_4^{3-}$ were measured at the 0.1 % sPAR depth, notably in the NATR/WTRA/SATL region. A P* of approximately 0.2 µmol $L^{-1}$ was measured at South Atlantic stations (Fig. 3c). A $NO_2^-$ concentration maximum was associated with the 1 % sPAR depth where an average of 117±112 nmol $L^{-1}$ was measured (Fig. 3e). Extremely low concentrations of $Si(OH)_4$ were measured in the well-lit surface ocean, progressively increasing at 1% and 0.1 % sPAR depths (Fig. 3f).

The dissolved oxygen concentration was generally within the range 150-280 µmol $L^{-1}$, with the exception of the 0.1 % sPAR depth in the NATR/WTRA/northern SATL, where concentrations approaching 100 µmol $L^{-1}$ were measured (Fig. 3d).

Plots of dissolved oxygen concentration against salinity (Fig. 4a), temperature (Fig. 4b) and $NO_3^-$ (Fig. 4c) are presented. Low dissolved oxygen concentrations measured at 0.1 % sPAR within NATR/WTRA/SATL provinces were associated with lower salinity, temperature, and relatively high $NO_3^-$. $R_{NO2}$ measurements associated with low-oxygen sampling depths were not anomalously elevated (Fig. 4d). In contrast, $R_{NO2}$ and physio-chemical characteristics of the eddy station identified as 39 °S distinguished it from the conditions associated with the majority of sampling stations within the transect.

### 3.3 Microbial cell distribution and abundance.

Profiles of chlorophyll-a concentration (Fig. 5a) indicated that within provinces of the North Atlantic, the sub-surface maximum generally deepened progressively south. The highest chlorophyll-a concentration was measured within the WTRA, although there was considerable variability in the depth of the chlorophyll-a maximum within this province. Within the South Atlantic, although sampling stations were classified as SATL province, chlorophyll-a profiles could be grouped into four illustrative sub-divisions, characterised by a general shallowing of the chlorophyll-a maximum progressively south.

Pico-eukaryotes (Fig. 5b) had a distribution that was similar to that of chlorophyll-a, in contrast to the profiles of the photosynthetic cyanobacteria *Synechococcus* and *Prochlorococus* (results presented in supplementary information Fig. S1). Bacterial cell distribution (Fig. 5c) was comparable between gyres, though less defined within the WTRA, and indicated that the highest bacterial abundance was measured within the relatively well-lit surface ocean.

### 3.4 Rates of $R_{NO2}$.

Within the physical, chemical and biological context described above, $R_{NO2}$ was measured (Fig. 6). For all $R_{NO2}$ data (Fig. 7a-c), the range of volumetric rate measurements was 0.02-13.27 nmol N $L^{-1}$ $d^{-1}$, n=88 triplicated measurements. When integrated from the surface to the 0.1% sPAR, the range of $R_{NO2}$ was 0.01-0.91 mmol N $m^{-2}$ $d^{-1}$ (average 0.11±0.08 mmol $m^{-2}$ $d^{-1}$). By extrapolating the vertically integrated results of each station (excluding 28°S and 39°S) from the equator into each hemisphere across a comparable distance (Fig. 8), the concentration of $NO_2^-$ regenerated was

calculated to be 0.36 mol N d$^{-1}$ (1 x 4457 km transect) and 0.72 mol N d$^{-1}$ (1 x 4637 km transect) for the North Atlantic Gyre and South Atlantic Gyre respectively. The box plot presented in Fig. 7d highlights variability within discrete sPAR bands of each hemisphere with mean R$_{NO2}$ values generally skewed towards the lower region of the data range. The distribution of rate measurements is presented in Fig. 7e while provincial variability, which averaged 25-fold, is presented in Fig. 7f. Using a 2 sample t-test assuming unequal variance, R$_{NO2}$ at 1% sPAR in the North Atlantic (M=0.76 nmol L$^{-1}$ d$^{-1}$, SD=0.67 nmol L$^{-1}$ d$^{-1}$) was significantly lower than R$_{NO2}$ at 1% sPAR in the South Atlantic (M=1.77 nmol L$^{-1}$ d$^{-1}$, SD=1.52 nmol L$^{-1}$ d$^{-1}$); t(15)=-2.40, p=0.03 (no significant difference between hemispheres was found for 14% and 0.1% sPAR depths). Volumetric rates of R$_{NO2}$ equated to a NO$_2$ turnover of 0.4-335 days (average 33 days).

**3.5 Multivariate analysis.**

Principle Component Analysis (PCA) of the full set of 24 environmental variables considered (Table 1; Fig. 9) clearly demonstrated that differences between sPAR bands were of the same magnitude or greater than differences along the transect length. Variation on PC1, accounting for 42.7 % of the total, was clearly associated with differences among sPAR bands. Differences among samples along the transect within sPAR bands were clearly associated with variation on PC2, accounting for 23.4% of the total. This was supported by 2-way ANOSIM with ordered factors, latitude and sPAR band, where the test for differences among latitudes removing differences over sPAR bands (R°=0.465, p<0.001) gave a lower value for the statistic than the corresponding test for differences among sPAR bands removing latitudinal trends (R°=1, the maximum possible, p<0.001). It was thus decided to analyse relationships among R$_{NO2}$ and environmental variables within each sPAR band separately.

Type 2 similarity profiles (SIMPROF) showed evidence for significant correlation structure among variables in all 3 sPAR bands (14% sPAR band, π=0.296, p<0.001; 1% sPAR band, π=0.205, p<0.001; 0.1% sPAR band, π=0.157, p<0.001). In the 14 % sPAR band there was a significant relationship ($\rho$=0.313, p=0.025) between patterns in R$_{NO2}$ and a subset of 3 explanatory variables consisting of light hours, chlorophyll-a and silicate. In the 1 % sPAR band the best subset, also of 3 variables, contained light hours and chlorophyll-a, but with NO$_2^-$ instead of silicate. The overall relationship was not significant ($\rho$=0.170, p=0.474). In the 0.1 % sPAR band the best subset ($\rho$=0.415, p<0.001) contained 6 variables, light hours and chlorophyll-a, temperature, NO$_2^-$, PO$_4^{3-}$ and P*. A 2-way search for the best overall match within sPAR bands removing any latitudinal effect selected light hours, chlorophyll-a and silicate, $\rho_{ave}$=0.267, p<0.001.

**3.6 Eddy features 28 °S and 39 °S.**

The majority of the AMT transect ran through quiescent waters free from rapidly varying mesoscale features. The equatorial belt is a region of counter-flowing currents, but these did not lead to any unusual features during this particular AMT campaign. However, anomalous profiles were observed at 28 °S and 39 °S (Fig. 1), in the form of conspicuously prominent R$_{NO2}$ (Fig. 7a-c). The spatial separation of CTD stations (~300 km) was too great to resolve these physical features. Thus, altimeter and Argo data were obtained subsequent to the cruise in order to provide context to these observations.

Mapped fields of Sea Surface Height (SSH) from CMEMS were used to calculate the mean sea surface topography and the variations about that (Fig. 10). A map of the root mean square variability (Fig. 10c) demonstrated that the typical changes north of 35 °S to be only ~0.1 m; however, even in this region there were a number of highs in the SSH anomaly (deviation from the mean) at the time of sampling the 28 °S station (Fig. 10a). The anticyclonic feature centred on station 28 °S had anticlockwise geostrophic flow along the contours in SSHA, whilst the total near-surface velocity recorded by the ADCP was mainly to the southwest. A series of SSHA composites a week apart showed a central anticyclonic feature to be moving westwards, whilst the contemporaneous Argo profiles showed that the thickest mixed layer was located close to the centre of this feature (Fig. 10b). A larger anticyclonic feature was seen 400 km further to the east. Examination of a time series of SSHA maps and Hovmöller diagrams (supplementary information Fig. S2 and S3) confirmed that these features had travelled from the Agulhas Retroflection, taking 2.5 years to get to this location.

The feature around CTD station 39 °S was very different. This is the region of the Brazil-Falklands Confluence, where two strong currents collide generating significant mesoscale activity and locally generated eddies. These features also move westwards, but their associated water masses were not from the Agulhas region. In this region the total

velocities from the ADCP agreed with the contour directions governing the geostrophic flow. As it is a much more
active region, Argo floats do not remain in the region for long, so the examples found (Fig. 10d) were more dispersed
than in the previous case. Using the previously stated definition of the depth of the surface layer produced much larger
values than at 28 °S, but again supported the concept that the surface layer was much deeper over the highs in SSH than
over the lows.

## 4. Discussion

### 4.1 Overview and objectives.

This study measured $R_{NO2}$ within the photic zone across an extended spatial scale (13,000 km), reporting significant
variability between and within provinces of the oligotrophic Atlantic Ocean between ~49 °N and ~39 °S. Insights add to
evidence that surface ocean nitrification is an important component of pelagic nitrogen cycling that modifies the
inorganic nitrogen inventory. Particular features of interest included a significant hemispheric difference in $R_{NO2}$ and
elevated $R_{NO2}$ associated with mesoscale eddy features. Analysis of potential links between $R_{NO2}$ and routinely
measured ecosystem variables indicated significant co-variability between $R_{NO2}$, chlorophyll-a concentration, the
duration of the light phase and silicate concentration, alluding to an association between $R_{NO2}$ and product(s) of
photosynthetic activity. However, a significant fraction (~35 %) of variability in $R_{NO2}$ remained unexplained, indicating
that additional factors were excluded from this analysis. We first consider the biogeochemical context to the study
before presenting rate observations and insights from statistical analysis.

### 4.2 Physical context.

A thermal range of ~17 °C typically existed in vertical 250 m water column profiles. While temperature has a direct
impact on rates of biological activity, the physical stability of the water column caused by thermal stratification
establishes spatially distinct biogeochemical niches (Giovanoni and Vergin, 2012). Within the well-lit surface ocean
that includes the 14 % sPAR, defined as the region between the sea-surface and the deep chlorophyll-a maximum
(DCM), turbulent mixing with nutrient rich deep water is supressed and biological activity maintains extremely low
inorganic nutrient concentrations. Relatively sharp physio-chemical gradients at depths similar to the 1 % sPAR create a
niche where low light-adapted phototrophs access relatively high inorganic nutrient concentrations, forming the DCM.
At greater depths, incorporating the 0.1 % sPAR and typically referred to as the twilight zone (approximately 100-
1000 m), water continues to cool. While there is relatively little information for the temperature sensitivity of
nitrification in the open ocean, Horak et al. (2017) demonstrated that a temperature effect on $NH_4^+$ oxidation was
observed at only two of four North Pacific stations. Horak et al. (2013) demonstrated a relative insensitivity to a
temperature range of 8-20 °C in their study of a fjord-like basin, although low pH at the study site may have constrained
any temperature response (Beman et al., 2011).

The range in salinity was ~3.2, with limits defined by notably fresher water in the sub-tropical region (WTRA), a
feature potentially related to intense rainfall within the inter-tropical convergence zone, and relatively high salinity in
the central Atlantic gyres. While a contributory role for salinity in shaping nitrifying community structure has been
identified in studies of estuarine sediments (Bernhard et al., 2010), such gradients are considerably more extreme than
those of the open oceans, where little, if any information is available to describe its influence upon variability in
nitrification rate.

The attenuation of light with depth leads to progressively decreasing daily light dose, ranging 2 orders of magnitude
across the three $R_{NO2}$ sampling depths (Fig. 2c). Superimposed upon diel and depth-related light variability was a
seasonal increase in day-light duration progressively south, associated with the transition from boreal autumn to austral
spring. Seasonally variable characteristics are evident in the photic zone of sub-tropical Atlantic gyres driven by
changes in solar insolation (McClain et al., 2004). Weaker solar insolation during winter supresses DCM productivity,
allowing higher nutrient concentrations to pass this 'biological filter' into the well-lit surface ocean. This mechanism
supports higher chlorophyll-a concentrations in the surface ocean (Taylor et al., 1986) and would presumably stimulate
broader biological activity, including the decomposition of recently produced organic matter and the regeneration of
nutrients. Seasonality in the rate of nitrification was not evident in the study of Yool et al. (2007), although such a
signal may exist (Newell et al., 2013).

Evidence demonstrates that light structures the distribution of nitrifying organisms, whose activity may be supressed within the photic zone (Olson et al., 1981; Shiozaki et al., 2016). Further, by poorly competing with photosynthetic plankton for $NH_4^+$ resource, nitrifier distribution and activity is indirectly constrained by the light regime (Smith et al., 2014; Ward, 1985). While the light/dark cycle may lead to a diel variability in nitrifying activity (increasing at night with the recovery from light inhibition; Smith et al., 2014), resource competition with phytoplankton may persist in the dark due to the photosynthetic cells' capacity for dark N-assimilation, especially of $NH_4^+$ (Clark et al., 2002).

### 4.3 Chemical context.

In terms of the chemical context to the study, characteristically low concentrations of $NO_3^-$, $PO_4^{3-}$ and $Si(OH)_4$ were evident within the well-lit surface ocean (14 % sPAR), progressively increasing at 1 % and 0.1 % sPAR (Fig. 3a and b and f). This distribution is consistent with biological draw-down, likely dominated by nutrient acquisition by phototrophs (Marañon et al., 2000; Poulton et al., 2006;). Extremely high concentrations were a prominent feature of nutrient profiles within the deep equatorial region.

In the mesopelagic ocean, nitrification maintains the accumulation of $NO_3^-$, while the diapycnal nutrient flux to the surface open ocean is relatively weak, especially during periods of low wind energy input. $NO_3^-$ supplied via nitrification in the photic zone is either extremely low (Clark et al., 2008; Newell et al., 2013) or below detection (Lomas et al., 2009). Alternative $NO_3^-$ and $PO_4^{3-}$ sources to the surface ocean includes introduction as a component of atmospheric dust (Baker and Jickells, 2017; Gruber and Sarmiento, 1997), subsurface injections via the action of internal tides over mid-ocean ridges (Tuerena et al., 2019) or the passage of mesoscale eddies (McGillicudy et al., 1998; Oschlies and Garcon, 1998). No direct correlation between nitrification and $NO_3^-$ concentration has been reported. A relationship with $PO_4^{3-}$ availability has been demonstrated in water treatment systems (de Vet et al., 2012), although there is no evidence to support this correlation in low nutrient environments.

Analytical capacity for nano-molar $NH_4^+$ measurements was unavailable for this study, creating a potentially significant shortfall in the data set, since $NH_4^+$ concentration has been demonstrated to influence the activity, distribution and composition of nitrifying communities (Bouskill et al., 2012; Smith, 2016). As the primary resource for nitrification, it is intuitive to expect a direct relationship between $R_{NO2}$ and $NH_4^+$ concentration (Smith et al., 2016), evidence for which supports the representation of this process as the specific rate ($d^{-1}$; Yool et al., 2007). However, this correlation is not consistently demonstrated (Bouskill et al., 2012) and the rate-resource relationship may not be linear, as implied by its representation as the specific rate (Martens-Habbena et al., 2009; Shiozaki et al., 2016).

A phosphate excess similar to 0.2 µmol $L^{-1}$ was evident at all depths in the WTRA and SATL (Fig. 3c). This feature is related to both source waters (Sarmiento et al., 2004) and the interhemispheric difference in N-fixation, supported by micro-nutrient inputs via atmospheric dust deposition (Baker and Jickells, 2017), leading to P* drawdown in the North Atlantic (Moore et al., 2009). There is no evidence of a direct relationship between P* and nitrification rate. However, in Atlantic regions of enhanced N-fixing activity, evidence suggests that organic material produced by diazotrophs is recycled within the surface ocean (Mulholland, 2007). Nitrification has rarely been measured simultaneously with N-fixation. Both were measured by Raes et al. (2020) and no relationship was reported, although it has been implied that N-fixation leads to an increase in the nitrate inventory (Mills et al., 2004). Nitrification likely has a role in redistributing newly fixed nitrogen, which is directly released as $NH_4^+$ or regenerated as $NH_4^+$ from the decomposition of released dissolved organic nitrogen (Newell et al., 2011). During this study, the seasonal low in dust deposition to the North Atlantic may have shifted resource constraints towards phosphate-Fe co-limitation, with implications for the extent and rate of N-fixation, and speculatively extending to the subsequent processes of N-remineralisation.

A Primary Nitrite Maximum (PNM) was evident over the entire transect (Fig. 3e). A range of 0.05-0.50 µmol $L^{-1}$ was measured, although sampling resolution may have limited the extent to which this feature was resolved (Beman, 2012; Meeder et al., 2012). Evidence suggests that nitrification has an important role in the formation and maintenance of the PNM against diffusive forces (Beman et al., 2012; Buchwald and Casciotti, 2013; Meeder et al., 2012; Newell et al., 2013; Peng et al., 2018; Santoro et al., 2013), with potential contributions from $NO_2^-$ release by phytoplankton (Beman et al., 2012; Santoro et al 2013). This study is unable to inform the debate surrounding PNM formation as the isotope dilution approach adopted here does not discriminate between the processes contributing to $R_{NO2}$ ($NH_4^+$ oxidation, algal $NO_2^-$ release and potentially a photolytic route which produces $NO_2^-$ from $NO_3^-$ (Zafiriou and True, 1979) and humic substances (Kieber and Seaton, 1999)), as discussed below. Across all depths, $R_{NO2}$ lead to an average

$NO_2^-$ turnover of 38.5 days (range 0.4-335 days) within the North Atlantic Gyre compared to an average of 37 days (range 2.9-160 days) for the South Atlantic Gyre. For the PNM, a range of 3-40 days was reported within the California current system (Santoro et al., 2013) and 33-178 days in the Arabian Sea (Buchwald and Casciotti, 2013).

The dissolved oxygen concentration was generally observed to be ~150-280 µmol $L^{-1}$, with variability linked to seawater temperature (Fig. 4b). Dissolved oxygen concentration has been shown to structure the activity and
distribution of nitrifying organisms (Beman et al., 2008; 2012; Ward, 2002), especially within low oxygen environments (Beman et al., 2008; Bouskill et al., 2012). A low oxygen feature of the 0.1 % sPAR equatorial upwelling water was distinguished in a series of correlation plots (Fig. 4a-c), linked to relatively low temperature and salinity in combination with a very high $NO_3^-$ concentration. This well documented oxygen minimum region is related to large scale circulation causing weak ventilation (Kartensen et al., 2008). This relative extreme in a combination of physico-
chemical conditions did not result in distinct elevations of $R_{NO2}$ (Fig. 4d). Similar results were noted by Shiozaki et al. (2016) in the North Pacific Ocean, where dissolved oxygen concentrations approaching 50 µmol $L^{-1}$ were not associated with an enhanced rate of nitrification. Nitrification within low-oxygen environments is of interest as it links N-inputs to marine environments via N-fixation to N-losses via denitrification and anaerobic ammonium oxidation (anammox), processes which are characteristically associated with marine regions of extremely low oxygen concentration (Francis et
al., 2007; Ward et al., 1989).

### 4.4 Biological context.

Profiles of biological parameters (Fig. 5 and Fig. S1) demonstrated apparent vertical structure. The chlorophyll-a distribution indicated that photosynthetic cells were present throughout the depth range used for $R_{NO2}$ measurements. The presence of measurable chlorophyll-a implied the potential production and release of dissolved organic matter, in
addition to organic matter released upon cell rupture (via grazing or viral attack). In contrast to the deep chlorophyll-a and picoplankton abundance maxima, bacterial cell profiles tended to describe higher abundance towards the well-lit region of the water column. Assuming that bacterial cell abundance reflected resource availability, the observed distribution implied the availability of photosynthetically derived labile dissolved organic nitrogen (DON) throughout the photic zone (noting that the DCM represents only a small fraction of water column productivity and biomass;
Marañon et al., 2000), and consequently the regeneration of $NH_4^+$. While the subsequent fate of regenerated $NH_4^+$ would relate to environmental factors that shape the open ocean ecosystem, molecular evidence supports the potential for nitrifying activity throughout the surface ocean (Church et al., 2010).

### 4.5 $R_{NO2}$ measurements within the study.

Isotope studies of the marine N-cycle are well established (Blackburn, 1979; Caperon et al., 1979; Ward, 2007;
2008) and the associated strategies each have benefits and risks. When applied to the pelagic oligotrophic ocean, the enrichment approach (detection of $NO_2^-$ isotopic enrichment following $^{15}NH_4^+$ addition and bottle incubation) has the advantage that enrichment can only be achieved via $NH_4^+$ oxidation. An associated risk is that $NH_4^+$ addition relieves resource limitation, leading to the measurement of 'potential' rates. Within open oceans, extremely low extant rates in combination with the need to achieve detectable enrichment of $NO_2^-$ often leads to the addition of relatively high $^{15}NH_4^+$
concentrations, 'carrier nitrogen' and/or prolonged incubation. In the present study we adopted the dilution approach (detecting the dilution of $^{15}NO_2^-$ due to $^{14}NO_2^-$ regeneration following bottle incubation) for the following reasons; (i) the analytical sensitivity of GCMS systems enabling isotopic analysis of inorganic nitrogen at nanomolar concentrations using short incubations (ii) the robust chemistry associated with Sudan-1 synthesis, as used for well-established colorometric analysis (iii) the synthetic nature of Sudan-1 which carries a low risk of sample contamination. The
disadvantage of this approach is that by measuring net $NO_2^-$ regeneration, the method does not discriminate between pelagic sources of $NO_2^-$, specifically $NH_4^+$ oxidation by nitrifiers and $NO_2^-$ release by photosynthetic plankton following intracellular $NO_3^-$ reduction. An issue common to all tracer approaches is the potential for inorganic nitrogen depletion within incubations due to microbial assimilation. We address these issues in relation to the dilution method below.

At 14 % sPAR, $R_{NO2}$ represented 25±19 % on average of the volumetric water column total at each station.
Photosynthetic cells were measured throughout the photic zone (Fig. 5a), reflecting the distribution of primary productivity which is evenly dispersed within the sunlit ocean and not constrained to the DCM (Poulton et al., 2006). This implied the potential for $NO_2^-$ release via incomplete $NO_3^-$ assimilation. $NO_2^-$ release from phytoplankton within the upper mixed layer has been reported in the California Current (Santoro et al., 2013), in the Red Sea during seasonal

transitions in water column structure (Al-Qutob et al., 2002; Meeder et al., 2012) and in the North Pacific (Wan et al., 2021). Under environmental conditions similar to those observed during this study for the upper mixed layer of the Pacific Ocean both Santoro et al., (2013) and Wan et al (2021) determined a limited number of rates of $NO_3^-$ reduction to $NO_2^-$ which are of the same order as those rates of $R_{NO2}$ reported here. As described above the method used to determine $R_{NO2}$ here does not differentiate between the two processes of ammonium oxidation and nitrate reduction and so there is potential for both processes to have contributed to the regeneration of $NO_2^-$ at this upper depth range. Wan et al., (2021) showed that $NO_3^-$ reduction by phytoplankton may supply in the order of $78 \pm 38\%$ of regenerated $NO_2^-$ to the upper mixed layer (Wan et al., 2021).

With greater depth, the 1 % sPAR was associated with sharp biogeochemical gradients where $R_{NO2}$ represented $44\pm23$ % on average of the volumetric water column total. The Primary Nitrite Maximum (PNM) was identified in close proximity to the 1 % sPAR depth; the persistence of this feature implies that it is formed and maintained by an imbalance between net biological production and consumption processes that exceed simultaneous rates of physical dispersion. Previous studies using direct rate observations and natural abundance stable isotope measurements have indicated that $NH_4^+$ oxidation is the major source of $NO_2^-$ at the PNM contributing approximately 70 to 90% of regenerated $NO_2^-$ (e.g. Buchwald and Casciotti, 2013; Santoro et al., 2013; Chen et al., 2021; Wan et al., 2021). Following $NO_2^-$ regeneration, the accumulation of $NO_2^-$ likely reflects a combination of photo-inhibition of $NO_2^-$ oxidising organisms in combination with light-limitation of $NO_2^-$ assimilation by photosynthetic organisms (Lomas and Lipschultz, 2006; Olson, 1981). $NO_2^-$ release derived from incomplete $NO_3^-$ assimilation contributes 7 to 15% of $NO_2^-$ to the PNM in the oligotrophic ocean. Direct observations within the California Current suggested that approximately 7 % of net $NO_2^-$ regeneration was derived from incomplete $NO_3^-$ assimilation (Santoro et al., 2013) whilst Wan et al., (2021) reported $10 \pm 6$ % for the North Pacific and Mackay et al (2011) indicated 10 – 15% for the Gulf of Aqaba. It may change with depth relative to the PNM feature (Al-Qutob et al., 2002; Beman et al., 2012; Lomas and Lipschultz, 2006; Mackey et al., 2011; Meeder et al., 2012; Newell et al., 2013; Santoro et al., 2013; Ward, 2005) and could be a source of variability in this study as the 1 % sampling depth was located either above, within or below the PNM feature on 15, 10 and 1 occasion(s) respectively (the remaining 7 stations could not be assigned due to a lack of PNM definition related to sampling resolution). Further, shoaling and deepening of the PNM by tens of meters over timescales of hours has been reported (Dore and Karl, 1996; French et al., 1983), linked to diel oscillation in the 1 % sPAR depth during the solar cycle as incident radiation intensity (and hence penetration) progressively increased towards solar noon and decreased towards sunset. Diel variability in the 1 % sPAR depth clearly takes place in tandem with the solar cycle in the Atlantic Ocean and is expected to influence biological processes associated with $R_{NO2}$. However, the extent of this influence has not been measured. $NO_2^-$ release by phytoplankton within the nitricline may be enhanced over relatively short temporal scales during such transitions (Lomas and Glibert, 1999; 2000).

The deepest samples were collected at 0.1 % sPAR where $R_{NO2}$ was $31\pm24$ % on average of the volumetric water column total. $NH_4^+$ oxidation activity at this depth is well documented; as depth increases below the DCM, the measurable yet substantial decrease in $NH_4^+$ oxidation rate is argued to reflect the progressively diminishing availability of labile particulate material settling out of the productive surface ocean (Smith et al., 2016; Ward, 2008). However, C-fixation by photosynthetic cells residing at the 0.1 % sPAR depth, either via photosynthetic or anaplerotic processes (Laws et al., 2014), has also been measured at an average rate of $35\pm22$ nmol C $L^{-1}$ $d^{-1}$ in the oligotrophic Atlantic Ocean (A. Poulton, pers. comm based on data in Poulton et al., 2006). Assuming Redfield stoichiometry, this would require a N-assimilation rate of $5.3\pm3.4$ nmol N $L^{-1}$ $d^{-1}$ (via inorganic or organic sources). The average rate of $R_{NO2}$ was $1.1\pm2.5$ nmol N $L^{-1}$ $d^{-1}$, implying that $R_{NO2}$ proceeded at a rate of comparable order to the phytoplankton N-demand required to support concomitant C-fixation. Photosynthetic cells utilise inorganic and organic forms of nitrogen to meet cellular demand; as inorganic forms require reduction to the level of $NH_4^+$ prior to assimilation, it is likely that reduced inorganic nitrogen (i.e. $NH_4^+$) and labile organic nitrogen would be utilised in preference to $NO_3^-$ under energy (light) limitation (Mackey et al., 2011).

The culmination of this evidence across sampling depths indicated that $NH_4^+$ oxidation likely dominated $R_{NO2}$ at 1 % and 0.1 % sPAR. $NO_2^-$ release by phytoplankton is likely to have contributed to $R_{NO2}$ at 14 % sPAR, with direct observations by others suggesting that this could be as high as ~78% (Wan et al., 2021).

Considering the potential for $NO_2^-$ assimilation by phytoplankton within incubation vessels, from the available data, ambient concentrations (directly measured by GCMS) during incubations were $5.7\pm2.2$ nmol $L^{-1}$ at 14 % sPAR, $5.3\pm2.5$ nmol $L^{-1}$ at 1 % sPAR, $5.3\pm2.0$ nmol $L^{-1}$ at 0.1 % sPAR. The associated change in $NO_2^-$ concentration during these

incubations was $-0.1\pm1.8$ nmol $L^{-1}$ at 14 % sPAR, $0.4\pm1.7$ nmol $L^{-1}$ at 1 % sPAR and $0.1\pm1.7$ nmol $L^{-1}$ at 0.1 % sPAR. $NO_2^-$ regeneration and consumption were approximately balanced in these incubations at all sPAR depths and there was no evidence of $NO_2^-$ depletion. During the balanced consumption of $NO_2^-$, sinks included $NO_2^-$ oxidation by nitrifying organisms and $NO_2^-$ assimilation by photosynthetic plankton. Data collected during the summer from the oligotrophic North Atlantic (Bay of Biscay, Clark et al., 2014) at 55 % sPAR indicated that at a $NO_2^-$ concentration of $28.7\pm1.7$ nmol $L^{-1}$ (i.e. sufficient $NO_2^-$ and light), $NO_2^-$ assimilation was $0.10\pm0.01$ nmol $L^{-1}$ $h^{-1}$. i.e. it was essentially insignificant. Consequently we do not anticipate that $NO_2^-$ assimilation represented an important sink at any depth, suggesting that $NO_2^-$ primarily continued through the nitrification pathway towards the regeneration of $NO_3^-$, a process that has been directly measured and found to be of similar order to rates presented here (Clark et al., 2008).

Within the physical, chemical and biological context described above (Fig. 6 and Fig. 7a-c), $NH_4^+$ was available to support $R_{NO2}$ in the range 0.02-13.27 nmol $L^{-1}$ $d^{-1}$. During a previous study of the Atlantic Ocean (Clark et al., 2008), 24-hour incubations incorporating a light/dark cycle returned an average $R_{NO2}$ value of $2.9\pm2.4$ nmol $L^{-1}$ $d^{-1}$ (15 stations, 2 photic zone depths). During the present study using 9-hour light phase incubations (a logistical compromise to enable high resolution sampling), an average $R_{NO2}$ value of $1.2\pm1.9$ nmol $L^{-1}$ $d^{-1}$ (32 stations, 3 photic zone depths) indicated broad agreement between the studies. The lower mean rates generated during this study (compared to Clark et al., 2008) may reflect either a recovery from light inhibition by nitrifiers during 24 hour light/dark incubations or more likely a different geographical extent. The 2008 study covering upwelling influenced waters in the east of the Atlantic basin and the current study transecting the open ocean at the centre of both gyres. The exclusion of a dark phase to the incubations used here is considered to offer only a minor effect due to the very low light environment experienced at both 1% and 0.1% sPAR.

The vertical distribution and magnitude of $R_{NO2}$ was consistent with previous observations, whereby low but detectable rates were measured within the well-lit ocean, a peak in $R_{NO2}$ was measured at 1 % sPAR, and a decline in rates was measured as depth increased below 1 % sPAR (Fig. 6, 7d and e, Al-Qutob et al., 2002; Beman et al., 2012; Newell et al., 2013; Peng et al., 2018; Santoro et al., 2013; Shiozaki et al., 2016). The causes for this distribution are a continuing subject of debate, but ultimately reflect a balance between multiple factors including the provision of labile organic matter supporting the regeneration of $NH_4^+$, competition for this resource from other microbes including photosynthetic cells, and grazing pressure by mixotrophs and heterotrophic protists upon the nitrifying community.

A striking feature of the $R_{NO2}$ data set was the 1 % sPAR rate range associated with South Atlantic stations (Fig. 7d), the mean of which was significantly greater than that measured in the North Atlantic Gyre. To illustrate this difference, extrapolating vertically integrated $R_{NO2}$ values across a comparable distance into each hemisphere from the equator, the amount of $NO_2^-$ regenerated was 0.36 mol N $d^{-1}$ over a 1 m x 4457 km distance north of the equator and 0.72 mol N $d^{-1}$ over a 1 m x 4637 km distance south of the equator. While the use of three sampling depths for this comparison represented low vertical resolution and was thus a limitation of the analysis, the average value of $0.11\pm0.08$ mmol $m^{-2}$ $d^{-1}$ is comparable to the value of $0.31\pm0.13$ mmol $m^{-2}$ $d^{-1}$ reported by the 6 integration depth study of Shiozaki et al. (2016) for the North Pacific Tropical Gyre.

We are unable to conclusively explain this novel distinction between hemispheres. However, it is recognised that the Atlantic gyres have persistent biogeochemical differences in dust deposition, the inorganic nutrient regime and biological features such as N-fixation (Baker and Jickells, 2017; Moore et al., 2009) that likely shape and influence broader biological activity within the surface ocean of each Atlantic gyre. Seasonality may be an additional component that contributed towards this hemispheric distinction. In their analysis of multiple AMT transects, Poulton et al. (2006) noted that high rates of carbon fixation at the sub-surface chlorophyll-a maximum characterizing SATL stations during October were potentially linked to the onset of spring conditions. At this time, nutrients supplied by winter mixing to the sub-surface chlorophyll-a maximum in combination with basin scale changes in irradiance, support the growth of larger eukaryotic phytoplankton (Letelier, 2004). The associated seasonal production of organic material would inevitably lead to an increase in decomposition and inorganic nutrient regeneration (including nitrification). Multivariate analysis, which identified significant co-variance between $R_{NO2}$ and chlorophyll-a, light duration and nutrients, provided support for this association, as discussed below.

Multivariate approaches were used to investigate potential structure within the data and to examine relationships between $R_{NO2}$ and environmental variables. However, it is first worth considering limitations of the data set used for this analysis. While there is strong evidence to suggest that $R_{NO2}$ reflected $NH_4^+$ oxidation activity rather than alternative

sources of $NO_2^-$, the measurement does not distinguish between archaeal or bacterial nitrifiers, which likely dominate separate niches within depth profiles (Beman et al., 2012). Nor does it consider the influence of changes in nitrifier community composition along the extended transect. Microbial cell abundance is no indication of the activity of specific microbial groups, or their importance to biological processes. Chlorophyll-a concentration does not reflect changes in species composition, or necessarily biomass, which are known to change in response to environmental
forcing within the Atlantic Ocean (Poulton et al., 2006). Within these limitations, the PCA ordination of the entire data set (Fig. 9) demonstrated greater similarity within a sampling depth, equating to horizontal scales of hundreds of kilometres, than between sampling depths equating to vertical scales of tens of meters. A similar result was previously observed (Poulton et al., 2017), implying that distinct niches existed between depth horizons.

Statistical analysis aimed to identify the subset of explanatory variables that, in combination, best matched the
observed pattern of $R_{NO2}$. This match was provided by a combination of chlorophyll-a concentration, light duration and silicate concentration. A similar outcome was reported by Shiozaki et al. (2016) in their study of the oligotrophic North Pacific, in which a correlation between the activity of photosynthetic cells and $NH_4^+$ oxidation rate was reported, although this relationship was not consistently observed. Chlorophyll-a and light duration as explanatory variables were common to all three sampling depths while depth-specific relationships incorporated additional variables. These results
implied that, either directly or indirectly, $R_{NO2}$ responded to a product of actively photosynthesising cells (as opposed to the activity of other microbes identified by AFC; see supplementary information Fig. S1). The production and release of labile dissolved organic matter during photosynthetic activity and its subsequent decomposition to regenerate $NH_4^+$ is an intuitive (and potentially spatially/temporally close) link between photosynthetic cells and nitrifying organisms that could be supported by this insight. However, a role for silicate has not previously been reported, potentially highlighting
a link to diatoms. Diatoms are relatively large, mineralising phytoplankton capable of rapid growth that contribute significantly to organic matter production and export in oligotrophic environments (Klass and Archer, 2006; Scharek et al., 1999). Alternatively, and perhaps more likely, silicate may be viewed as a tracer for sporadic nutrient inputs from below the nutricline (to include $NO_3^-$, $PO_4^{3-}$) that stimulate short-term growth of photosynthetic cells in oligotrophic environments (Poulton et al., 2006). In contrast to other resources supplied by vertical mixing (e.g. $NO_3^-$, $PO_4^{3-}$), silicate
is only utilised by diatoms which represent a fraction of the microbial community. It is reasonable to expect that following a localised mixing event, elevated silicate concentrations persist for longer than the simultaneously introduced $NO_3^-$ and $PO_4^{3-}$, which are drawn down due to localised increases in primary productivity. Silicate effectively becomes a tracer for water that hosts the products of active or recently enhanced productivity, with subsequent decomposition and nutrient regeneration.

The current view of low nutrient environments sees a dominant role for the picoplankton community which supports the microbial food web, with the sporadic growth of large phytoplankton following episodic nutrient inputs due to localised short-term physical instability (Dandonneau et al., 2003; Poulton et al., 2006; Scharek et al., 1999). Increased nutrient availability favours larger and faster growing cells that outcompete the picoplankton to transiently dominate the microbial community in open ocean environments (Kiørbe, 1993). Organic matter released during such transient events
(directly from living cells or following cell lysis) would support a short-term localised increase in N-regenerating activity, including $R_{NO2}$. Evidence indicates that archaea, which frequently dominate the euphotic zone nitrifying community, possess a high resource affinity and respond rapidly to $NH_4^+$ pulses in culture (Beman et al., 2012; Martens-Habbena et al., 2009). Whether the statistically significant link between $R_{NO2}$ and a combination of chlorophyll-a, light duration and silicate relates to diatoms specifically (which have an absolute requirement for silicate and are obligate
phototrophs i.e. they do not express mixotrophy unlike most other photosynthetic plankton, a nutritional mode that affords flexibility in resource limited environments and decreases dependence upon photosynthetic $CO_2$ fixation; Mitra et al., 2014) or more likely relates to a broader stimulation of transient micro-plankton growth remains to be tested. Should nitrifying activity be responding to a chain of events in the very recent past, instantaneous rates of nitrification may not be fully explained by routine measurements of extant environmental conditions, such as those used in the
present study. A detailed examination of labile DON production and decomposition leading to the regeneration of $NH_4^+$ may be required in combination with studies of nitrification.

The prerequisite for $NH_4^+$ to support $R_{NO2}$ implies the availability of a (semi-)labile source of DON. While this may be generated locally, as speculated above, an alternative (or potentially complimentary) hypothesis would be that DON is supplied by Eastern Boundary Upwelling Ecosystems (EBUE; Mahaffey et al., 2004; Torres-Valdés et al., 2009).
Letscher et al. (2013) measured a global average [DON] in surface waters (<50 m) of 4.4 µmol $L^{-1}$, although elevated concentrations were measured adjacent to, and downstream of major upwelling systems. Zonal westward gradients

across the Atlantic gyres implied a sink for DON. A mechanism that potentially explained this distribution invoked seasonal mixing dynamics at the Atlantic's eastern flanks which subduct DON-enriched water from EBUE's that subsequently penetrates the gyre over time scales of months to years. This water is encapsulated within a density horizon that constrains it to the deep euphotic zone. During simulated mixing experiments, the rate of DON decomposition in this water was reported to be threefold higher than for surface waters (Letscher et al., 2013), potentially reflecting differences in the microbial community composition (Treusch et al., 2009), and their associated ability to access semi-labile DON. We speculate that the enhanced values of $R_{NO2}$ associated with the 1 % sPAR of both Atlantic gyres (Fig. 7d) is supported, to some extent, by DON laterally advected from the upwelling systems of the Atlantic's eastern flank. We further speculate that the hemispheric distinction in $R_{NO2}$ values reflects the higher [DON] supplied by the Benguela compared to the Mauritania upwelling system (Letscher et al., 2013). Should this be the case, it would imply that the remineralisation of EBUE derived DON represents a new N source to the gyre interior, potentially supporting primary and export production. The retrieval of DON-N leaves a relatively C-enriched organic molecule progressively more resilient to microbial decomposition in a region of large scale downwelling (i.e. C-export via the microbial carbon pump; Jiao et al., 2010). It remains to be tested whether EBUE derived DON is remineralised across extended spatial scales within sub-tropical gyres, a mechanism that could operate in both the Atlantic and Pacific Oceans, and what the broader implications are for pelagic biogeochemistry.

Conspicuously prominent $R_{NO2}$ were associated with stations 28 °S and 39 °S, with measured ranges of 1.5-8.6 nmol $L^{-1}$ $d^{-1}$ and 1.3-13.3 nmol $L^{-1}$ $d^{-1}$ respectively (Fig. 4d, 7a-c). Selected biogeochemical characteristics further distinguished station 39 °S (high P*, $O_2$ concentration, lower salinity and temperature; Fig. 3 and 4) although those of station 28 °S were typical of the province. The use of Earth Observation data (sea surface temperature, ocean colour) revealed no anomalous features co-incident with these stations. However, Sea Surface Height Anomaly (SSHA) data evidenced that 28 °S and 39 °S were associated with mesoscale eddy features at the point of sampling (Fig. 9). Analysis revealed that no other stations of this study coincided with discernible mesoscale eddy features at the point of sampling.

Eddy 39 °S was locally generated. In contrast, eddy 28 °S likely originated within the Agulhas Retroflection, traversing and potentially influencing South Atlantic Gyre biogeochemistry before dissipating on the Brazilian shelf some 3.5 years later (Fig. S2 and S3). Nencioli et al. (2018) tracked similar features, showing that they provided a coherent transport of water masses, preserving some of the physical and biological signals over more than two years. The vertical flux of inorganic nutrients induced by the dynamics of traversing eddies has been argued to balance the Sargasso Sea nutrient budget (McGillicuddy et al., 1998). Similarly, the $NO_3^-$ deficit observed between new production requirements and diffusive $NO_3^-$ supplies in the Atlantic Gyres may be reconciled to some extent by eddy dynamics (Planas et al., 1999). While the nutrient regime of 28 °S was comparable to the provincial average, elevated $R_{NO2}$ (some 2-6 times higher than the provincial average) implied unusual biological activity. Such activity is not sustained in isolation, but must couple (over appropriate scales of space/time) to the activity of other biological processes (e.g. nutrient acquisition by phototrophs, photosynthetic growth, DON production and release, decomposition and $NH_4^+$ regeneration, $NO_2^-$ oxidation), presumably supported by new nutrients and/or semi-labile EBUE derived DON physically introduced to the well-lit water column by eddy dynamics. Unfortunately, given that only 3 $R_{NO2}$ observations were made at station 28 °S, located at the edge of a feature which measured approximately 300 km in diameter, little can be said about how representative of the feature this activity is, the relevance to broader nitrogen cycling activity or its significance as a source of atmospheric $N_2O$ (Dore and Karl, 1996).

### 4.6 Conclusions.

Biogeochemical models typically apply a specific nitrification rate of 0.02 to 0.10 $d^{-1}$ (Moore et al., 2002; Wang et al., 2006; Christian et al., 2002; Jiang et al., 2003), comparable to the value of 0.011-0.113 $d^{-1}$ derived from the present study assuming 10-100 nmol $L^{-1}$ $NH_4^+$, a range consistent with historical measurements (Clark et al., 2008; British Oceanographic Data Centre data archives; www. BODC.ac.uk). Through statistical analysis we demonstrated that ~65% of variability within $R_{NO2}$ was explained through co-variability (not correlation) with environmental variables, identifying a role for chlorophyll-a concentration, light duration and silicate. We speculate that silicate is a tracer for seawater that hosts the products of active or recently enhanced phytoplankton growth, with subsequent organic matter decomposition, $NH_4^+$ regeneration and hence nitrification. This leads us to conclude that a role for DON, and specifically the short-term production, release and decomposition of labile material is amongst the factors contributing to the unexplained variability in $R_{NO2}$ (notwithstanding the omission of $NH_4^+$ concentration form this study). Such relationships may need further exploration in both observational and modelling studies.

A novel interhemispheric distinction in $R_{NO2}$ was observed, but not adequately explained by our hypothesis based on statistical analysis. While seasonality may contribute towards this feature, we invoke the influence of the Atlantic's EBUE's as a potential explanation, which deliver a higher concentration of DON to the southern compared to the northern gyre interior. Such activity would reflect an influence of Atlantic EBUE's at significantly extended spatial scales and simultaneously represent the delivery of new nitrogen to the photic zone and the transformation of dissolved organic matter towards recalcitrant forms.

**Data availability**

All AMT 19 metadata can be downloaded from the British Oceanographic Data Centre website: https://www.bodc.ac.uk/data/hosted_data_systems/amt/, in addition to $R_{NO2}$ data (doi:10.5285/56df6379-5b2b-160b-e053-6c86abc01f07) and flow cytometry data (doi:10.5285/a2104adc-e994-6789-e053-6c86abc0d557).

**Author contribution**

DRC – designed experiments, undertook sample and data analysis, written manuscript; APR – conducted N-cycle fieldwork investigations, contributed to manuscript preparation; CMF - conducted N-cycle fieldwork investigations; CH - conducted inorganic nutrient analysis of fieldwork samples; LA – mass spectrometry sample analysis; PJS - statistical analysis, contributed to manuscript preparation; GDQ - satellite data analysis, contributed to manuscript preparation; SG – satellite data analysis; GL –data analysis, contributed to manuscript preparation; GT- conducted flow cytometry analysis of fieldwork samples, contributed to manuscript preparation.

**Competing interests**

The authors declare that they have no conflict of interest.

**Acknowledgments**

We thank the crew of the AMT19 cruise. MSLA data were obtained and analysed through the NERC Earth Observation Data Acquisition and Analysis Service (NEODAAS), and further data were provided by the European Space Agency's Sea Level CCI. This is contribution number 320 of the AMT program, which was also supported by the Natural Environment Research Council funded Microbial Carbon Pump project (Clark Co-I; NE/R011087/1). We acknowledge funding from the UKRI through the National Capability Long-term Single Centre Science Programme, Climate Linked Atlantic Sector Science (Grant no. NE/R015953/1). We also thank Dennis Hansell and two anonymous reviewers for constructive and insightful comments.

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

**Figure 1.** Map of the study area, indicating the position of sampling stations for N-cycle studies (red dots), the approximate transect distance and the associated Longhurst provinces (Longhurst, 1998); North Atlantic Drift (NADR), North Atlantic subtropical gyre (NAST), North Atlantic tropical gyre (NATR), Western tropical Atlantic (WTRA), South Atlantic gyre (SATL). The location of two eddy features is also indicated.

**Figure 2.** Contour plots of water column temperature (a) and salinity (b) measured during CTD rosette casts. The calculated daily light dose (PAR intensity x light phase duration) received at sampling depths is presented (c), in addition to the duration of the light phase (in hours). The approximate position of Longhurst provinces (defined in Fig. 1) is indicated. Symbols represent 14 % sPAR (circle), 1 % sPAR (triangle), 0.1 % sPAR (square). The location of two eddy features is indicated by vertical dashed (28 °S) and dotted (39 °S) lines.

**Figure 3.** The concentration of nitrate (a), phosphate (b), phosphate excess (P*; c), oxygen (d), nitrite (e) and silicate (f). The location of two eddy features is indicated by vertical dashed (28 °S) and dotted (39 °S) lines. The approximate location of Longhurst provinces (defined in Fig. 1) is indicated. Symbols represent 14 % sPAR (circle), 1 % sPAR (triangle), 0.1 % sPAR (square). The location of two eddy features is indicated by vertical dashed (28 °S) and dotted (39 °S) lines.

**Figure 4.** Plots of dissolved oxygen concentration vs salinity (a), temperature (b) and $NO_3^-$ (c) within Longhurst provinces (defined in Fig. 1). Open circles represent 0.1 % sPAR data, highlighting distinct characteristics associated with specific NATR/WTRA/SATL sampling points. To represent the distribution of rate data, the 88 $R_{NO2}$ measurements were arranged in ascending order (d, insert has linear scale) with an indication of sampling depth, or the association with a specific feature (low oxygen, defined as < 150 µmol $L^{-1}$, or mesoscale eddy). Measurements within eddy features tend towards the high end of the scale, as do measurements at 1 % sPAR. Low oxygen measurements show no clear distributional trend while 14 % and 0.1 % sPAR values tend towards the low end of the scale.

**Figure 5.** Profiles of chlorophyll-a concentration (a), the abundance of pico-eukaryotes (b) and bacteria (c) within Longhurst provinces (defined in Fig. 1). Data are presented as a ratio of the maximum value measured for each parameter throughout the transect. Profiles frequently describe comparable distributions within a defined province. However, data is sub-divided within the SATL province to illustrate apparent distinctions in vertical structure.

**Figure 6.** $R_{NO2}$ measurements at 7 °S within the context of selected physical, chemical and biological measurements, illustrating relationships between water column characteristics.

**Figure 7.** Volumetric rates of $NO_2^-$ regeneration at 14 % (a), 1 % (b) and 0.1 % (c) sPAR. Error bars represent two standard deviations of triplicate measurements. The approximate position of Longhurst provinces (defined in Fig. 1) is identified in these plots. The box plot (d) presents data for each hemisphere (excluding 28 °S/39 °S), as sPAR depths. The 10th, 25th, 75th and 90th data percentiles are presented with data median (line within box). A histogram of the rate distribution and frequency is presented in panel (e). An indication of $R_{NO2}$ variability is presented in panel (f), as the number of minimum value multiples that equate to the largest value within each province.

**Figure 8.** $R_{NO2}$ integrated to the 0.1 % sPAR depth at individual stations. These values (excluding stations 28 °S and 39 °S) are extrapolated between stations extending from the equator to a comparable distance north (33 °N) and south (35 °S) of the equator. The concentration of ammonium-N oxidised was calculated to be 0.36 mol N $d^{-1}$ over a 1 m x 4457 km distance north of the equator and 0.72 mol N $d^{-1}$ over a 1 m x 4637 km distance south of the equator. Seawater temperature (as presented in Fig. 2) is included to provide illustrative context to the data. The approximate position of Longhurst provinces (defined in Fig. 1) is indicated.

**Figure 9**. Principle component analysis (PCA) ordination, highlighting the clustering of samples from specific light equivalent depths. Inter-sample variation on PC1 is associated with differences among the sPAR bands, and explained 42.7 % of the total. Inter-sample differences along PC2 reflect latitudinal variation along the transect and explained 23.4 %. In combination, ~65 % of variability was explained, leaving ~35 % unexplained.

**Figure 10**. Altimetry, ship and Argo profiler data relating to the two anomalous stations, indicated by magenta dots with black boundaries. (a) Sea surface height anomaly (SSHA) at the time of the 28 °S station. The altimetry shows a high (anticyclone) centred on the station, with a larger feature further east. The arrows provide the currents at 54 m

depth from the shipborne ADCP. (b) The same altimetry data as contours (intervals of 0.03 m), with dots indicating Argo profiles within 5 days of ship passage, and the numbers being the Mixed Layer Depth (depth at which temperature drops more than 3°C below the surface value. (c) The root mean square variability in SSHA showing the pattern of large signals associated with the Brazil-Falklands Confluence, and the position of the 39°S station. (d) Contours of SSH (0.10 m increment, with red for the zero and positive values; blue lines for negative values). The arrows show currents from the shipborne ADCP at 54 m depth, and the dots and green numbers indicate Argo profiles and the depth of the surface layer within 10 days of ship transit. (Green dashed lines link successive profiles of the same float, with all three being advected eastward).

**Table 1**

Results of Principle Components analysis. Eigenvalues are presented in panel (A). In panel (B), Eigenvectors are presented, the coefficients in linear combinations of variables making up principal components (PC).

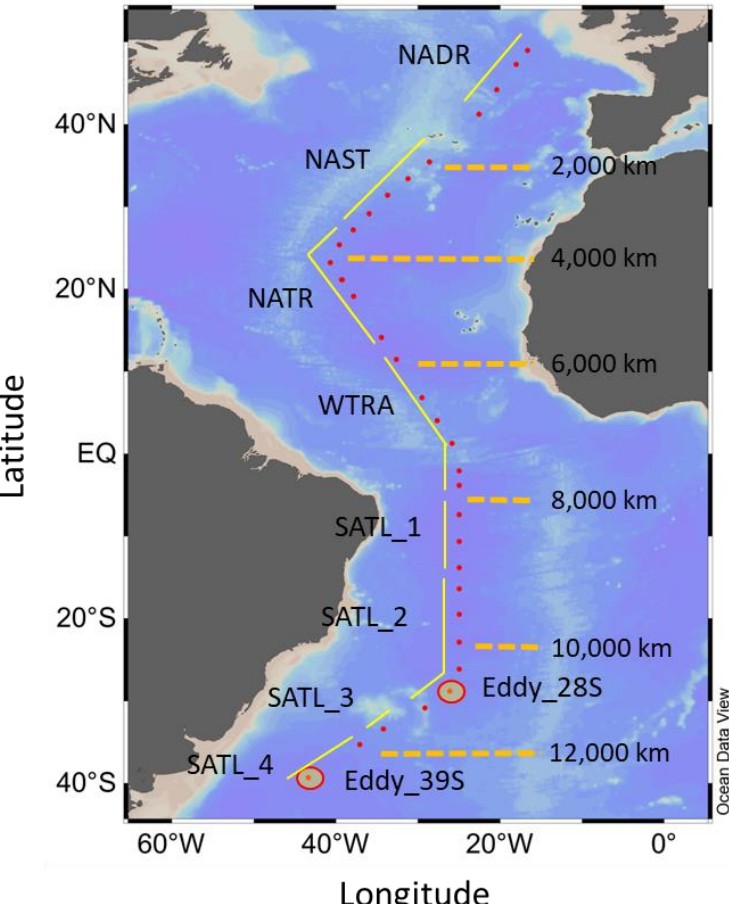

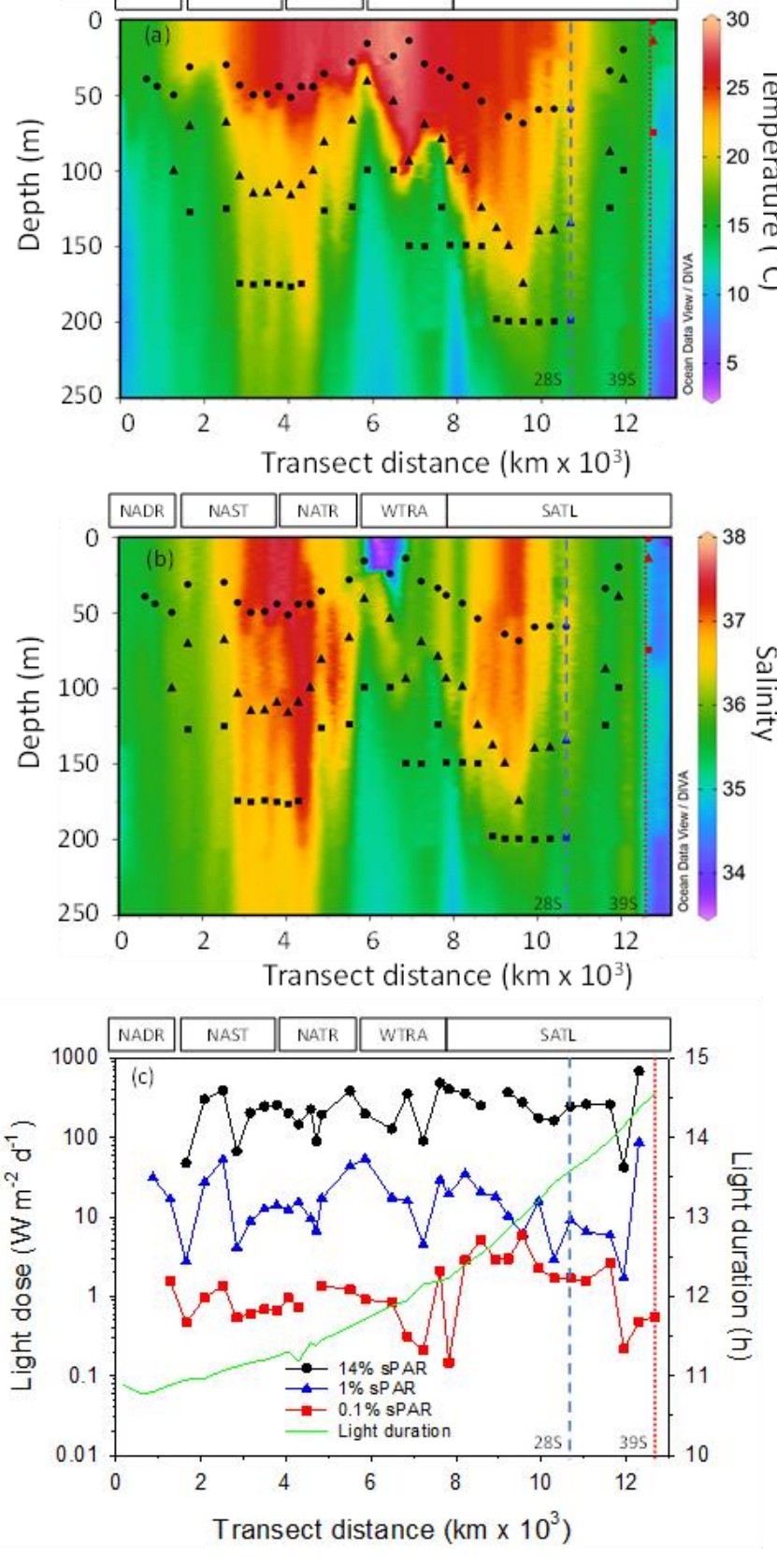

985

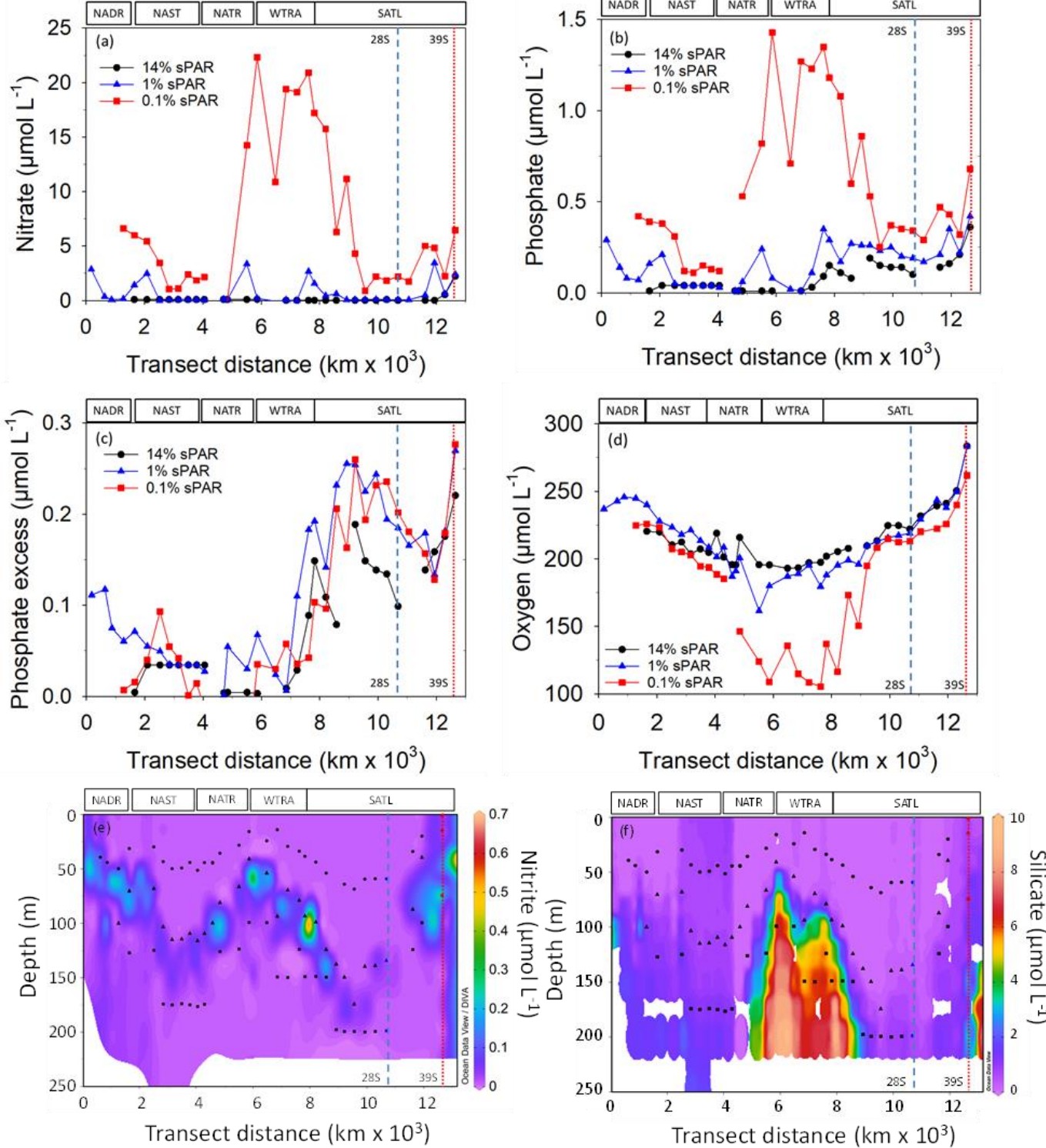

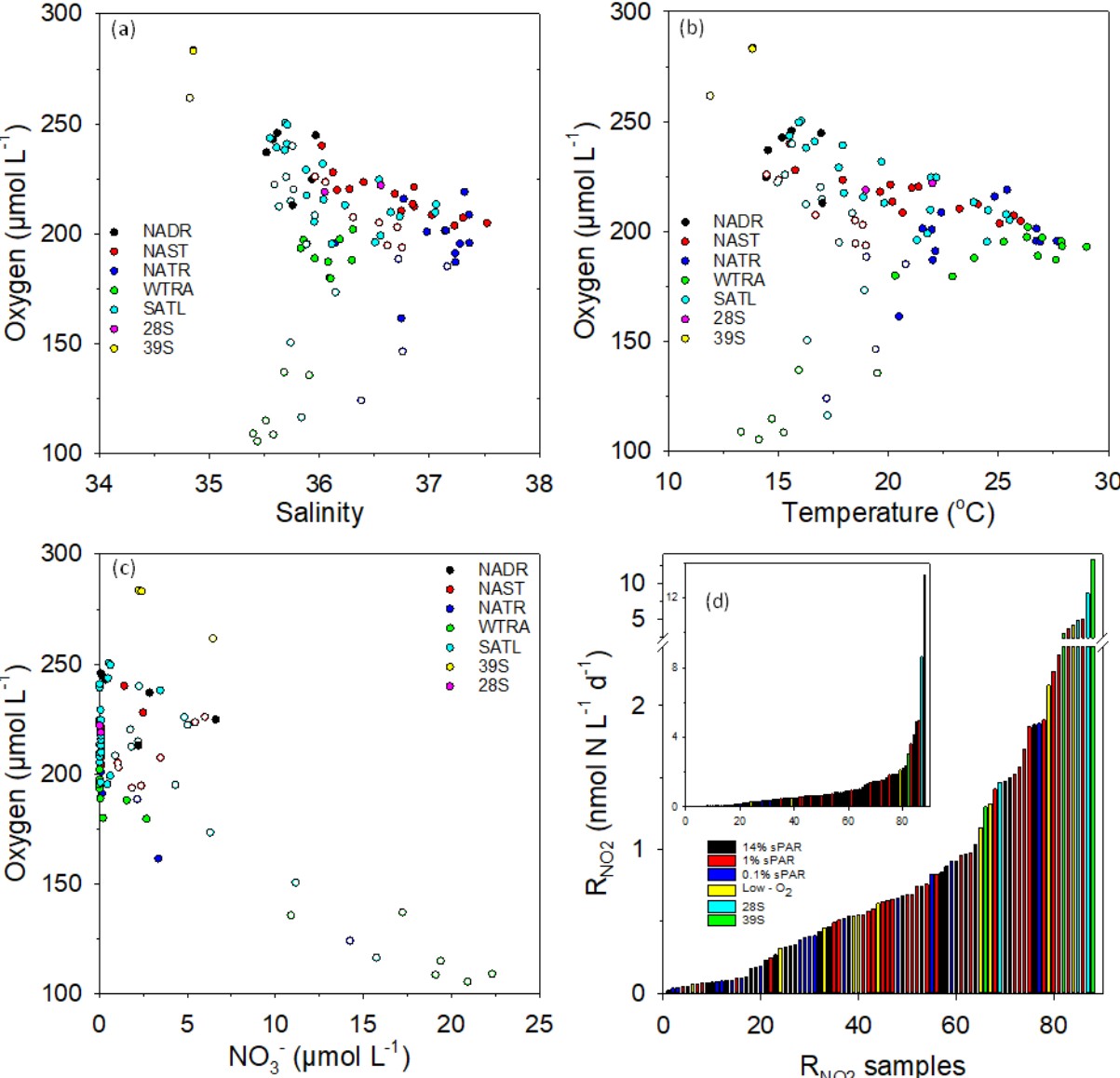

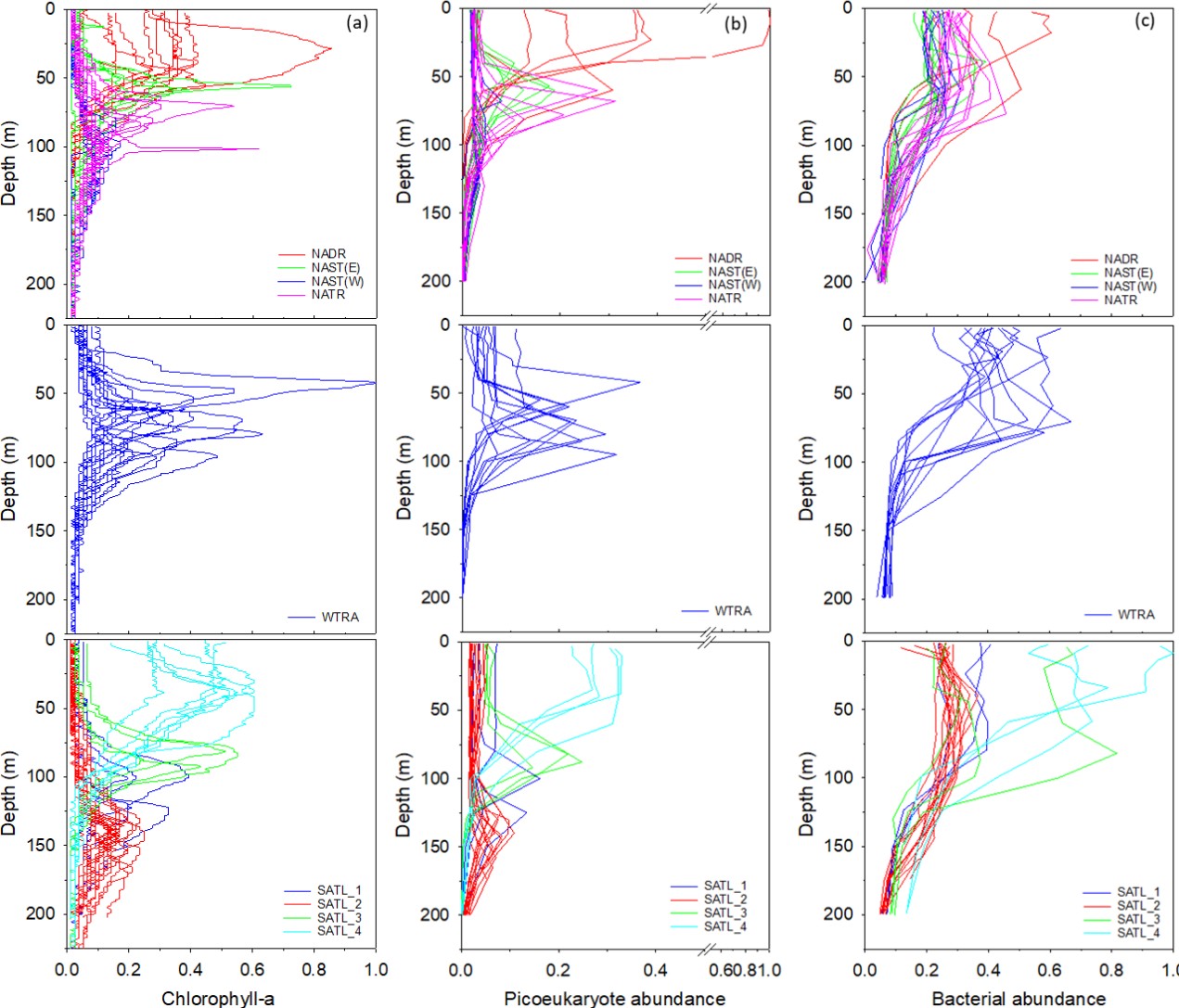

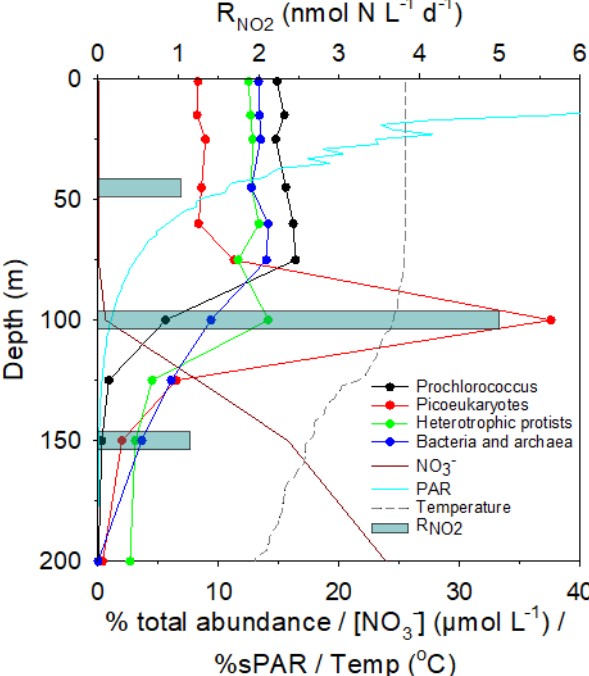

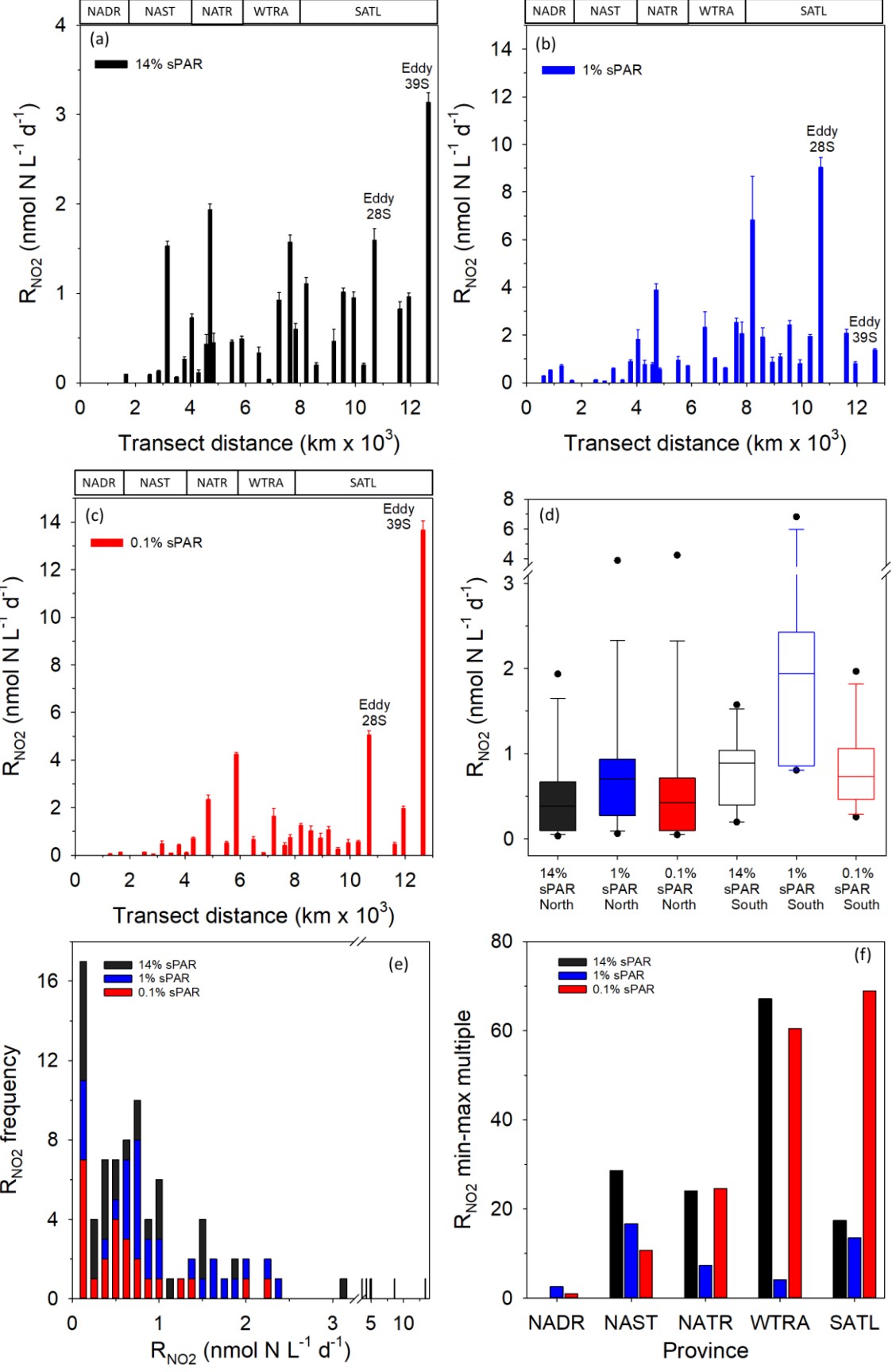

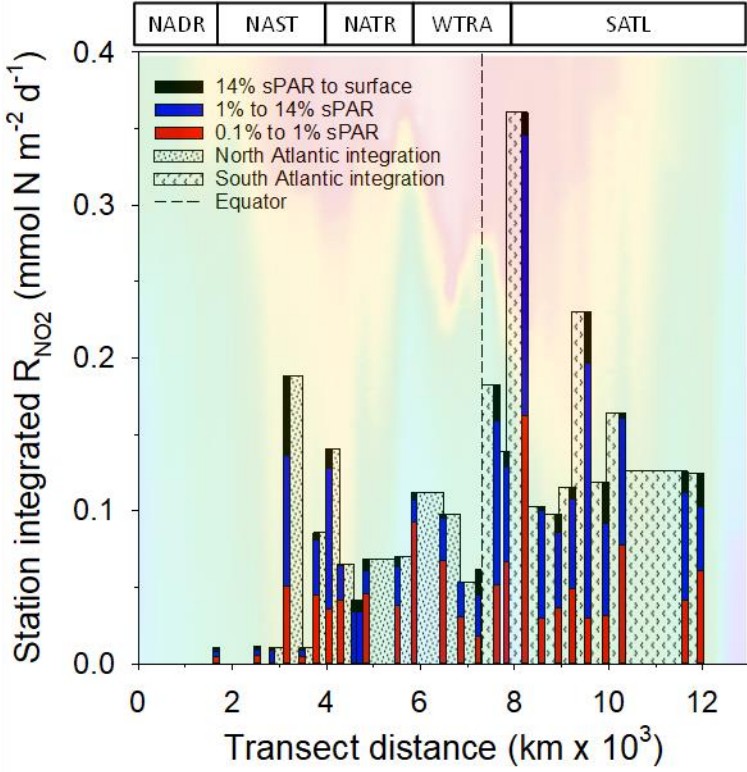

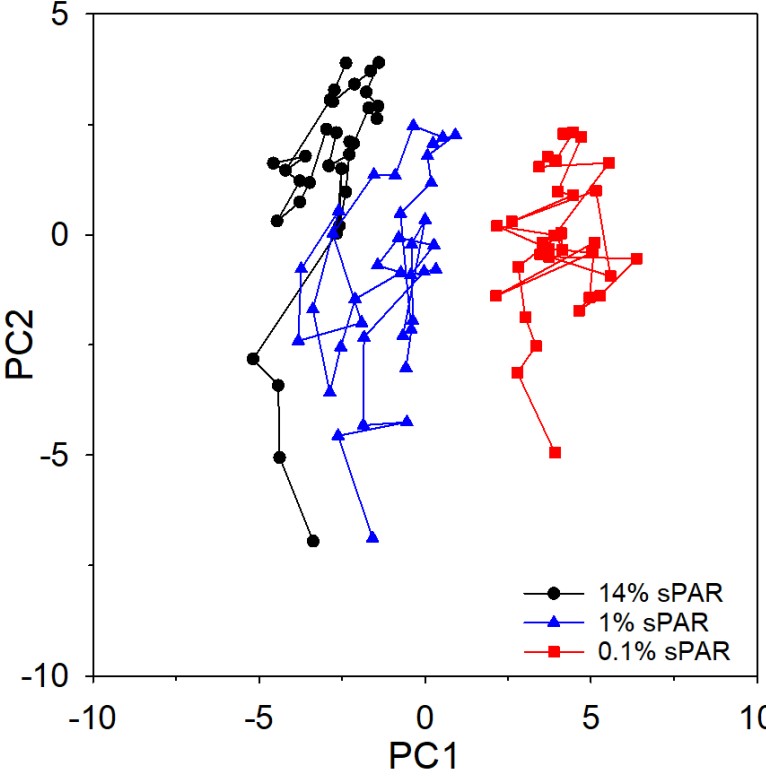

1010

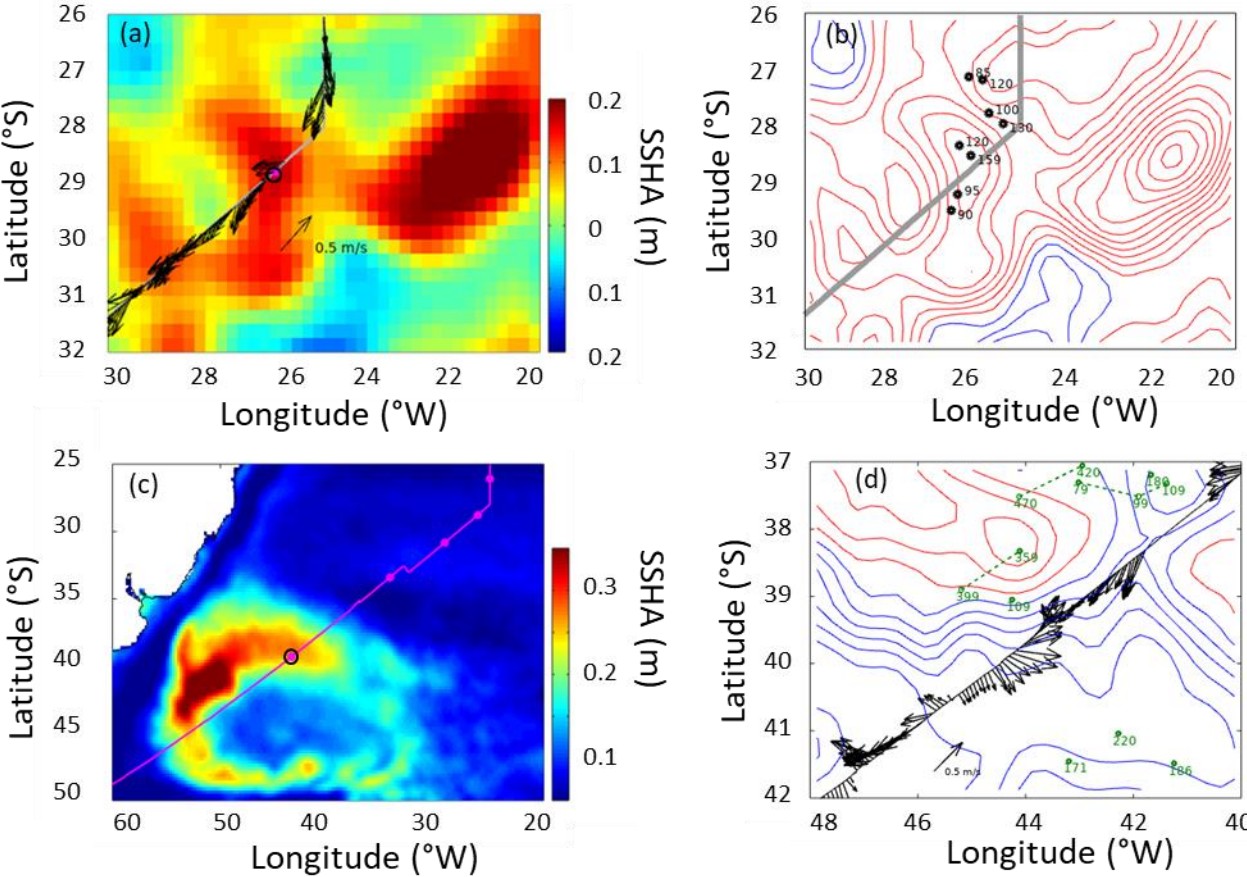

1015

1020

1025

1030

A

| PC | Eigenvalues | %Variation | Cum.%Variation |
|---|---|---|---|
| 1 | 9.81 | 42.7 | 42.7 |
| 2 | 5.38 | 23.4 | 66.1 |
| 3 | 1.64 | 7.1 | 73.2 |
| 4 | 1.19 | 5.2 | 78.3 |
| 5 | 1.16 | 5.1 | 83.4 |

B

| Variable | PC1 | PC2 | PC3 | PC4 | PC5 |
|---|---|---|---|---|---|
| Par ($\mu$W cm$^{-2}$) | -0.279 | 0.097 | -0.105 | 0.143 | -0.039 |
| Light (hours) | -0.037 | -0.267 | -0.449 | 0.171 | 0.249 |
| Light (W m$^{-2}$ d$^{-1}$) | -0.28 | 0.086 | -0.123 | 0.15 | -0.03 |
| Conductivity (S m$^{-1}$) | -0.185 | 0.298 | 0.118 | 0.054 | 0.244 |
| Chlorophyll (mg m$^{-3}$) | -0.056 | -0.288 | 0.258 | -0.04 | 0.332 |
| Dissolved oxygen ($\mu$mol L$^{-1}$) | -0.151 | -0.076 | -0.373 | -0.413 | -0.379 |
| Salinity | -0.036 | 0.307 | 0.152 | -0.283 | 0.164 |
| Temperature (°C) | -0.196 | 0.279 | 0.104 | 0.084 | 0.26 |
| NO$_2^-$ ($\mu$mol L$^{-1}$) | -0.003 | -0.252 | 0.263 | -0.466 | -0.136 |
| NO$_2^-$ + NO$_3^-$ ($\mu$mol L$^{-1}$) | 0.259 | -0.165 | 0.162 | 0.138 | -0.034 |
| NO$_3^-$ ($\mu$mol L$^{-1}$) | 0.244 | -0.169 | 0.223 | 0.2 | -0.103 |
| PO$_4^{3-}$ ($\mu$mol L$^{-1}$) | 0.203 | -0.243 | -0.072 | 0.122 | 0.266 |
| Si ($\mu$mol L$^{-1}$) | 0.236 | -0.066 | 0.165 | 0.254 | -0.047 |
| P* ($\mu$mol L$^{-1}$) | 0.013 | -0.221 | -0.398 | -0.232 | 0.499 |
| *Synechococcus* (cell mL$^{-1}$) | -0.255 | -0.152 | -0.046 | 0.227 | -0.26 |
| *Prochlorococcus* (cell mL$^{-1}$) | -0.25 | 0.146 | 0.206 | -0.09 | 0.166 |
| Pico-eukaryote (<2$\mu$m; cell mL$^{-1}$) | -0.231 | -0.197 | 0.24 | -0.247 | 0.067 |
| Nano-eukaryote (2-12$\mu$m; cell mL$^{-1}$) | -0.261 | -0.2 | 0.027 | 0.008 | 0.075 |
| Coccolithophores (cell mL$^{-1}$) | -0.205 | -0.174 | 0.194 | 0.149 | 0.19 |
| Cryptophytes (cell mL$^{-1}$) | -0.089 | -0.311 | 0.191 | -0.184 | -0.022 |
| Low nucleic acid bacteria (cell mL$^{-1}$) | -0.279 | -0.14 | 0.015 | 0.158 | -0.124 |
| High nucleic acid bacteria (cell mL$^{-1}$) | -0.292 | -0.099 | 0.043 | 0.187 | -0.083 |
| Heterotrophic protists (cell mL$^{-1}$) | -0.239 | -0.226 | 0.057 | 0.143 | -0.107 |