# Peer review of "Nitrite regeneration in the oligotrophic Atlantic Ocean"

_Biogeosciences, 2021_

## Referee Comment (RC1)

**Comments on "Nitrification in the oligotrophic Atlantic Ocean"**

**General comments:**

This study presents a high horizontal resolution dataset of ammonia oxidation (AO) rate over a 13, 000 km transect of the Atlantic Ocean. The dataset could be valuable to improve our understanding of the interaction between AO and environmental factors in the open ocean, and to facilitate the oriented-development of ocean biogeochemical models. However, the methodology, structure of this manuscript, the presentation of the results, and the Discussion need to be further condensed and reorganized. The comments are as follows:

**Specific comments:**

1. By measuring the NO2− concentration and nitrogen isotope before and after the incubation, the dilution method of obtaining the ammonia oxidation rate contains some issues worthwhile discussing:there are some potential influencing factors for the NO2− product that are not carefully discussed in the manuscript. For example, the incomplete assimilatory reduction of nitrate by phytoplankton can produce $NO_2^-$ especially in light condition, probably resulting in AO overestimation in this study. On the other hand, $NO_2^-$ consumption term during the incubation can be derived, e.g., oxidation by nitrite oxidizing microorganisms and assimilation by phytoplankton. (2) The equations for calculating the AO rate were not provided in the manuscript.

2. Pity that ammonium concentration, which is one of the key factors controlling ammonia oxidation rate in open ocean, was not measured to achieve their goal of "clarifying the relationship between the AO rate and environmental factors".

3. The water-column-integrated AO rate that the authors used to explore the difference of AO rate between northern and southern hemispheres, was only three-water-layer based. The vertical resolution of this data was not enough to address the scientific

questions.

4. The literature survey was insufficient, some highly relevant literatures were not included, resulting in misleading statements. e.g. Line ~295: "there is relatively little information for the temperature sensitivity of nitrification in the open ocean", references like Baer 2014; Horak et al., 2017; Zheng et al., 2020 were recommended; Line ~315: "Evidence demonstrates that light structures the distribution of nitrifying organisms, whose activity may be suppressed within the photic zone (Shiozaki et al., 2016; Olson et al., 1981)", Xu et al. (2019) gave a thorough discussion on light effect on ammonium oxidation"; Line ~410: When the paper talks about the vertical distribution of ammonia oxidation, the work by Wan et al. (2018) should be mentioned. Wan et al. (2018) found the nitrification was low or undetectable in the mixed layer and increased rapidly downward around the nitracline. And the depth of the nitracline represents a robust spatial boundary between ammonium assimilators and oxidizers in the stratified ocean.

**Technical corrections:**

1. Line 9: The product of ammonia oxidation is nitrite, not nitrate.

2. Line 58-60: "Evidence identifies roles for light, $NH_4^+$ concentration, organic matter flux and competitive interactions with phytoplankton (Fawcett et al., 2015; Shiozaki et al., 2016; Olson, 1981; Smith et al., 2014; 2016; Bresseler and Boyd, 2009; Newell et al., 2011)." is suggested to change it to "Evidence identifies roles for light (Olson, 1981; Shiozaki et al., 2016; Xu et al., 2019), $NH_4^+$ concentration (Smith et al., 2016; Xu et al., 2019), organic matter flux (Ward and Zafiriou, 1988; Newell et al., 2011) and competitive interactions with phytoplankton (Fawcett et al., 2015; Smith et al., 2014)." Moreover, the paper Buesseler and Boyd (2009) focused on processes that control particle export and flux attenuation in the open ocean, not relevant to nitrification.

3. Line 190-199: To make a clear description of results, please match the figure to the corresponding text, e.g. "The dissolved oxygen concentration ,,,, where

concentrations approaching 100 μmol L$^{-1}$ were measured (Fig. 3d)".

4. In Figure 3a-d, squares stand for samples from 1% sPAR depths, while in other figures (2 and e-f) squares represent 0.1% sPAR.

5. Line 214: show the profiles of cyanobacteria *Synechococcus* and *Prochlorococus* or delete ", in contrast to the profiles of the photosynthetic cyanobacteria *Synechococcus* and *Prochlorococus* (results not shown)".

6. Keep the format of all the secondary headings consistent. For example, "3.3. Microbial cell distribution and abundance." ended with a period, while secondary headings of "3.4., 4.2.-4.4." did not have periods.

7. Line 219: Figure 6 did not show the vertical profile of ammonium oxidation rate; The rate unit in the text is nmol L$^{-1}$d$^{-1}$, different from nmol L$^{-1}$h$^{-1}$ in the Figure 7. There were two Figure 7f in the figure 7 (Line 830).

8. Line 223-224: What's the meaning of "the concentration of ammonium-N oxidized"; Provide the full name of "NAG" and "SAG".

9. Line 234 and Figure 9 would be misleading when describe the PC2 as "representing latitudinal variation", PCn just representing the combinations of the linear transformations of variables without explicit meaning. Furthermore, it is hard to infer the implications of PC1 and PC2 from 2-way ANOSIM results.

10. The result 3.6 and Figure 10 can be moved to supplementary information. And the title of color bar of Figure 10 (a) and (c) should be add on the figure.

11. Line 264-265: The Hovmöller diagrams should be shown to prove the conclusion that "these features had travelled from the Agulhas Retroflection, taking 2.5 years to get to this location."

12. Line 275-285: Delete the paragraph of 4.1 "Overview and objectives" or move to introduction.

13. Line 315-320: This paragraph is more than summarized the light effect on nitrifier without linking to their own data.

14. Tremendous typos and mistakes, e.g., Line 323: "Fig. 3.a, b, f" should be "Fig. 3a, b, f"; Line 364: "(Kieber and Seaton, 1999))" should be "(Kieber and Seaton, 1999)"; Line 372: "Fig. 4d" should be "Fig. 4b"; Line 373: "(Ward, 2002, Beman

et al., 2008; 2012)" should be "(Ward, 2002; Beman et al., 2008, 2012)"; Line 447: "Shiozaki et al., (2016)" should be "Shiozaki et al. (2016)"; Line 484: "(Fig. 4d; 7a-c), " should be "(Fig. 4d, 7a-c)"; Line 493: "Nencioli et al., (2018)" should be "Nencioli et al. (2018)"…

15. Line 409: "a decline in rates was measured as depth increased below 1 % sPAR (Fig. 6; 7d, e;", there is no rates data in Fig. 6.

16. Line 509-510: "Considering all rate data, results implied that more active nitrogen cycling took place in the South compared to the North Atlantic Gyre." This conclusion may not true since the $NH_4^+$ concentrations in South and North Atlantic Gyre may be different.

17. Nearly the same sentences occur repeatedly, e.g., Line 221-224 vs Line 417-419; description about labile dissolved organic matter (Line 453-455 vs 520-524)…

Ref:

B.B. Ward, O.C. Zafiriou (1988). Nitrification and nitric oxide in the oxygen minimum of the eastern tropical North Pacific, Deep Sea Research Part A. Oceanographic Research Papers, 35(7), 1127-1142.

Zhenzhen Zheng, Liwei Zheng, Min Nina Xu, Ehui Tan, David A. Hutchins, Wenchao Deng, Yao Zhang, Dalin Shi, Minhan Dai, Shuh-Ji Kao (2020). Substrate regulation leads to differential responses of microbial ammonia-oxidizing communities to ocean warming. Nature Communications, 11, 3511.

Min Nina Xu, Xiaolin Li, Dalin Shi, Yao Zhang, Minhan Dai, Tao Huang, Patricia M Glibert, Shuh-Ji Kao (2019). Coupled effect of substrate and light on assimilation and oxidation of regenerated nitrogen in the euphotic ocean. Limnology and Oceanography, 64(3), 1270-1283.

Wan, X.S., Sheng, H.-X., Dai, M., Zhang, Y., Shi, D., Trull, T.W., Zhu, Y., Lomas, M.W., Kao, S.-J. (2018). Ambient nitrate switches the ammonium consumption pathway in the euphotic ocean. Nature communications 9, 915.

---

## Referee Comment (RC2)

**Reviewer report**

Peer review report on "Nitrification in the oligotrophic Atlantic Ocean"

**1. General comments**

The manuscript 'Nitrification in the oligotrophic Atlantic Ocean' by Clark et al. investigated the $NO_2^-$ production rate in the euphotic zone across a large area of the Atlantic ocean. Additional environmental variables were presented to examine the potential controls on this key process in the sunlit ocean.

They found $NO_2^-$ production was active at all depths, and across the study areas with large spatial variability, a striking feature of rate difference between the two hemispheres was highlighted. The authors identified that Chl-a, duration of the light phase, and Si concentration best explained the observed rates.

While such large-scale observation on the biogeochemical rate is encouraging and inspiring, there are several key issues that need to be further clarified and resolved before drawing the conclusions presented in the manuscript:

1.1 Methodology

Instead of using the $15NH_4^+$ labeling and dark (or paired light-dark) incubation, the authors chose the $15NO_2^-$ dilution method, and the incubation was carried out under light conditions, this inevitably induces several issues:

1) The measured rate stands for the total $NO_2^-$ production rate rather than ammonia oxidation rate. Specifically, in the sunlit ocean, it would be expected that phytoplankton also play a significant role in producing $NO_2^-$, particularly in the upper euphotic zone where $NO_2^-$ production might be dominated by phytoplankton rather than nitrifier. For this reason, I would suggest re-consider the measured rate as '$NO_2^-$ production rate' rather than 'nitrification rate.'

2) The lack of rates been measured in dark conditions would also cause a significant bias of $NO_2^-$ production rates in the sunlit ocean as both phytoplankton and nitrifiers are sensitive to light.

3) I am also concern about the potential effect of $NO_2^-$ consumption by phytoplankton in the nutrient-deprived surface ocean on the rate measurement and calculation.

1.2 Result and discussion

1) The lack of in-situ NH4+ measurement is another major flaw of the current study. The substrate concentration is one of the most fundamental controls for ammonia oxidation in the marine environment, so it would be of great relevance in understanding the environmental control on this process. Is that possible for the authors to

2) The main finding—a significant hemispheric difference of the NO2- production rate is exciting. More detailed information and discussion are encouraged further to explore the potential mechanism of this novel finding. Either it is attributed to spatial or seasonal variation, the production, and remineralization of organic matters might be a fundamental control. Thus, I suggest expanding the discussion on this point to examine this hypothesis, i.e., to examine the productivity (satellite, Bio-Argo, literature…), the strength of remineralization, etc.

3) The authors declared that NO2- production was enhanced in the mesoscale eddies. While I appreciate that mesoscale processes exert profound influences on marine biogeochemistry, more evidence is expected to bolster this argument. For instance, mesoscale eddy is a ubiquitous feature in the global ocean, have the authors examined whether they sampled in other eddies aside from these two eddies reported here? Can the authors expand a bit to examine the underlying causes? i.e., Increased primary production? Intensified remineralization? Alleviated competition between nitrifier and phytoplankton etc.

4) I am also curious about the effect of light, i.e., any explanation for the lack of correlation between the measured rate vs. light intensity, but the duration of light can well explain the observed rates? Also, it is counterintuitive to see the positive correlation between light duration and NO2- production rate.

**2. Specific comments**

Line 1: You do not need the full stop at the end of the tile.

Lines 26-27: Why not NH4+?

Lines 91-95: I encourage the authors to expand to clarify the method, particularly the detection limit and the accuracy of the trace level of nutrient measurement.

Line 106: What criteria are used to define the euphotic zone, 1% PAR or 0.1 PAR%? )

Line 109: Taken the extremely low level of NO2- outside the PNM layer (i.e., less than 10nM in many samples?), is that mean that only <1nM of 15N-NO2- (<10%) was added into the incubation system? Would that cause bias if the trace amount of NO2- was assimilated by the

phytoplankton and/or bring a significant challenge in measuring the NO2- isotope at such low concentration?

Line 116: What is the reason for performing a ten h day-time incubation? Taking the inhibition of ammonia oxidizers by light in the sunlit ocean and their ability to recover under dark, ten-hour day-time incubation rather than a full day incubation covering the light-dark cycle would cause underestimating ammonia oxidation rate. In another aspect, NO2- release during assimilatory NO3- reduction appears to be another key source of NO2- in the euphotic zone. The dilution method used in this study cannot distinguish these two sources of NO2- and thus would overestimate the ammonia oxidation rate.

Lines 120-128: The method used here for NO2- isotope analysis is quite complex that involves multiple steps and reagents. Extreme care should be taken to avoid any contamination for the isotope analysis of the trace amount of NO2-. For this reason, some introductions on the detection limit and accuracy will benefit the audience on the accuracy of the result.

Lines 135-139: Similarly, I suggest further clarifying the calculation process and detection limit of the rate.

Lines 175-176: Citation required?

Lines 187-189: Perhaps some description on the integrated light intensity (dose)? And the potential reason for the absence of a corresponding increase of irradiance?

Lines 213-214: Any reason for not showing the results of *Synchronous* and *Prochlorococcus*? Given that these cyanobacteria are the most representative and abundant phytoplankton in the oligotrophic ocean and the main competitors for NH4+, information on the distribution of *Synchronous* and *Prochlorococcus* should be relevant of interpreting the observed ammonia oxidation rate.

Lines 220-221: Since the vertical resolution is relatively coarse, it would not be robust to do the depth-integrated rate calculation.

Lines 305-308: Although there is a clear spatial pattern of the light duration of the investigated area, the light dose did not show a similar pattern. Additional analysis between the rates and light dose might also be helpful.

Lines 318-320: Numerous studies have discussed the effect of light and the competition between nitrifiers and phytoplankton in the sunlit ocean. It deserves to discuss further the potential effect of the diel cycle and the role of phytoplankton on consuming and /or producing NO2-.

Lines 336-342: Regretfully, that NH4+ concentration was not measured in this study! The substrate acts as the most critical factor in regulating the ammonia oxidation rate in the ocean. The significant correlation between substrate concentration and ammonia oxidation rate is more often reported in the literature, so it would be better to point it out. Also, the NH4+ concentration is usually very low (i.e., lower than the reported Ks of the ammonia oxidizers) in the oligotrophic gyres; saturation kinetics appears to be very unlikely in the study area.

Line 361-362 This is another key issue of the study. A main weakness of the dilution method lies in its unavailability of discerning the potential sources. It is very likely that the rate of NO2- production by ammonia oxidizers in the euphotic zone is overestimated, particularly at the upper euphotic zone where nitrifiers are inhibited by light and outcompeted by the phytoplankton, the NO2- production rate measured here is more likely attributed to phytoplankton rather nitrifiers.

Lines 385-386: See my comment above. I suggest showing the profiles of *Synchronous* and *Prochlorococcus*.

Lines 386-387: Doesn't that mean the potential of NO2- released by phytoplankton might occur throughout the sampling depth and thus should be taken into account in the discussion?

Lines 415-430: It is interesting to see the systematic rate difference between the hemispheres. The authors have proposed several potential causes for explaining the phenomenon. Nevertheless, I would like to see more detailed information on bolstering these hypotheses, i.e., does the productivity/ biomass (either from Argo or satellite) show a similar spatial pattern between the hemispheres during the cruise? Counterintuitively, the duration of light increases from the northern hemisphere to the southern hemisphere. As light is an inhibitor for the nitrifiers, is there any explanation for the apparent positive correlation between ammonia oxidation rate and light duration?

Lines 434-435: What does this sentence mean?

Lines 471-472: Do the bacterial abundance indicate higher remineralization rates in the South hemisphere?

Lines 483-505: It is also quite interesting to see the significantly enhanced rate in 28°S and 39°S, which was attributed to the mesoscale process. While it deserves to expand a bit more to discuss it: 1) it is true that the prominent high rate occurred in these two eddies, worth noting that mesoscale eddy is a ubiquitous feature in the ocean, are there any other eddies at the sampling stations during the cruise? If so, is there any evidence of stimulation of ammonia oxidation by the eddy? 2) Is that possible to look into the biomass, productivity,

heterotrophic activities, directly or indirectly, to bolster the argument that the high rate was due to the eddy?

Line 795: I suggest plotting the depth of the mixed layer in Fig. 2a.

Line 805: I suggest presenting nitrite and silicate (Fig. 3e and 3f) in the same way with Fig. 3a-3d, because the vertical resolution (only three depths) is too coarse for the extrapolation.

Line 825: How was the NO3- profile derived, given that only three depths were sampled for nutrient analysis in each station?

Line 830: Some statistical analysis (i.e., if the difference between the hemispheres is significant) would further bolster the difference between the two hemispheres in Fig. 7d.

Line 840: Again, the depth-integrated rate in Fig. 8 was based on the rates measured at three depths, which was insufficient for extrapolation.

---

## Author Comment (AC1)

[Figure]

**Figure 6.** $R_{NO2}$ measurements at 7 $^0$S within the context of selected physical, chemical and biological measurements, illustrating relationships between water column characteristics.

5

---

## Author Comment (AC2)

We thank the reviewers for their comments. To provide context, this manuscript is the latest in a series of studies in which we have investigated the regeneration of inorganic nitrogen in the marine environment. Following the publication of the isotope dilution methods (Marine Chemistry 2007, 2008, which provided full details of the strengths, weaknesses, isotope equations, practical detection limits, precision and sensitivity) we investigated $NH_4^+$ regeneration and nitrification in the oligotrophic Atlantic Ocean (Clark et al, L&O 2008). This was the first basin-scale study of N-regeneration in the Atlantic. Its data contributed to a global modelling study of nitrification (Yool et al, Nature, 2007). Following similar investigations in the Iberian upwelling (Clark et al L&O, 2011), the Mauritanian upwelling (Clark et al BGS, , 2016) and the European Shelf Sea (Clark et al BGS, 2014), we returned to investigate the Atlantic Ocean with higher resolution (the present study). This study has identified features of N-cycling in the Atlantic Ocean that have not previously been reported. It therefore contains novel insights into the biogeochemistry of a globally important microbial process at the basin scale. This data has contributed to a recently constructed global database of marine nitrification measurements to support a new modelling study of marine nitrification in collaboration with the Geophysical Fluid Dynamics Laboratory and the Ward Research Laboratory, both at Princeton University (USA). Finally, we note that having posted the BGS pre-print of this paper on ResearchGate, it received > 68 reads within 4 weeks. This demonstrates a great deal of interest in the subject area.

We first address general comments made by both reviewers, before addressing specific comments raised by individual reviewers.

**Opening remarks [OR] Points raised by both reviewers:**
Reviewers commented on the isotope dilution method.
RC1 '…the methodology….condensed and re-organised…';
**[OR-1]**

Measuring nanomolar inorganic nutrient concentrations in marine seawater is challenging – measuring biological rate processes at nanomolar nutrient concentrations is extremely challenging. Such measurements are still not routinely made and so there is relatively little information about N-regeneration in open oceans. Consequently, there is a clear knowledge gap to fill. We considered the various approaches to the measurement of nitrogen regeneration, including their advantages and limitations when we originally published the method used here (Clark et al 2007). By referring to this paper, which presents full analytical details and equations for data analysis, we have condensed the methods section appropriately. The key features of this method are:

- sensitivity – the method directly measures N-flux at nanomolar DIN concentrations, without the use of inhibitors, C-N conversion factors or 'carrier' N to satisfy detection limitations of MS systems. The GCMS system is orders of magnitude more sensitive than IRMS systems.
- the ability to use _genuinely trace_ isotopic additions to incubations in contrast to isotope enrichment approaches which must use multiple- or order of magnitude-increases in ambient [DIN] within open oceans in order to measure a rate that is significantly greater than analytic error (due to the extremely low rates of N-regeneration in open oceans).
- the method avoids directly stimulating the process to be measured, unlike the isotopic enrichment approach. Due to the addition of excess $^{15}NH_4^+$, the enrichment approach provides an estimation of the 'potential' ammonium oxidation rate, rather than an estimation of the actual rate. This is a serious shortfall of the isotope enrichment approach in open oceans, as it releases the process of interest from any substrate limitation, providing a measure that is unrelated to the environmental constraints from which the sample was derived.
- ability to separate sample collection from land-based analysis thus maximising ship-based observational capacity.

Actions for revision
- While the MS is within BGS length guidelines, we will add to the Material and Methods section, outlining key information about the method to include LOD, sensitivity, equations.

RC1 stated 'the dilution method contains some issues…' which were also outlined by RC2;

- 2) The lack of rates been measured in dark conditions would also cause a significant bias of NO2- production rates in the sunlit ocean as both phytoplankton and nitrifiers are sensitive to light. >

**[OR-2]**

During our previous study of the Atlantic Ocean (Clark et al 2008) we used 24 hour incubations, and measured an average hourly $R_{NO2}$ rate of 119.3±100.1 pmol.L$^{-1}$,hr$^{-1}$ (15 station, 2 photic zone depths). During the present study, the average hourly rate $R_{NO2}$ rate 50.5±78.7 pmol.L$^{-1}$.hr$^{-1}$ (32 stations, 3 photic zone depths). i.e. they are comparable. In addition, the use of 24 hour incubations to incorporate a full light/dark cycle brings with it additional complications associated with biological rate measurements using stable isotopes. As incubation time increases, nitrogen tracer becomes distributed between fractions (dissolved, particulate and potentially gaseous), reflecting the multiple oxidation states and biologically essential processes that nitrogen is involved with. The potential recovery of isotope for analysis diminishes with time, decreasing sensitivity. Consequently, such incubations often use the shortest practical duration, in order to avoid these consequences. While light is an important modulating factor (as stated in the MS), it does not lead to complete inhibition of nitrification in the photic zone (as evidenced by multiple studies). It may be anticipated that equal or higher rates would be achieved in the dark, although our rates over both time frames are comparable. If anything, our measurements define a conservative lower boundary.

Actions for revision

- We will include this direct comparison of average hourly rate derived from 9 and 24 hour incubations to alleviate the reviewers concern.

RC2 stated;

- 3) I am also concern about the potential effect of NO2- consumption by phytoplankton in the nutrient-deprived surface ocean on the rate measurement and calculation.

**[OR-3]**

The method directly measures the $NO_2^-$ concentration before and after the incubation. This information is used for rate calculations - it is not derived through calculation (as suggested by RC1). From the available data, ambient concentrations during incubations were 5.69±2.24 nmol.L$^{-1}$ at 14% sPAR, 5.27±2.45 nmol.L$^{-1}$ at 1% sPAR, 5.29±2.02 nmol.L$^{-1}$ at 0.1% sPAR. The associated change in $NO_2^-$ concentration during these incubations was -0.10±1.82 nmol.L$^{-1}$ at 14% sPAR, 0.39±1.66 nmol.L$^{-1}$ at 1% sPAR and 0.09±1.68 nmol.L$^{-1}$ at 0.1% sPAR. From these measurements we can state that (i) $NO_2^-$ was not depleted during the period of incubation e.g. due to $NO_2^-$ assimilation and (ii) the regeneration of $NO_2^-$ (via $NH_4^+$ oxidation) was approximately balanced by $NO_2^-$ consumption (i.e. $NO_2^-$ oxidation) . The assimilation of $NO_2^-$ is rarely measured. However, from our previous investigation of the European Shelf surrounding the UK during the summer (Clark et al 2014) we note that in this more productive system (in which biological rates will generally be higher), the rate of $NO_2^-$ assimilation was < 5% of simultaneously measured $NO_3^-$ assimilation. At an ambient $NO_2^-$ concentration of 8.9 nmol.L$^{-1}$ (i.e. almost double the average concentration measured in the oligotrophic Atlantic) the simultaneously measured rate of $NO_2^-$ assimilation was 0.008±0.002 nmol.L$^{-1}$.h$^{-1}$. At double this concentration, 19.2 nmol.L$^{-1}$, the rate of $NO_2^-$ assimilation was 0.10±0.02 nmol.L$^{-1}$.h$^{-1}$. i.e. it was essentially insignificant. Consequently, in the oligotrophic ocean, we do not anticipate that $NO_2^-$ assimilation will represent a significant sink at any depth.

Actions for revision

- We will include a condensed version of this discussion point in a revision.

RC1 commented on the lack of $NH_4^+$ concentration data while RC2 stated;

- The lack of in-situ NH4+ measurement is another 'major flaw' of the current study.

**[OR-4]**

Points are introduced prior to this comment by RC2 as 'key issues', not 'flaws'. The lack of $NH_4^+$ concentration data was indeed a frustrating shortfall of the data availability. It was beyond our ability to influence during the cruise planning stage (i.e. this was not a careless oversight). It reflected limitations associated with scientific resources – the

measurement of nanomolar $NH_4^+$ is labour-intensive. Within the oligotrophic gyres, values are < 20 nmol/L, almost invariably (as evidenced by historical data archived at BODC by PML), with higher concentrations only associated with fringing regions, or those influenced by upwelling (EBUE or equatorial). Consequently, the effort for nutrient analysis was focused upon NO2/NO3/PO4/Si. However, it is without doubt that this study contributes to knowledge regarding the environmental factors that influence ammonium oxidation rates in the open ocean. The absence of this data did not stop us explaining > 65 % of the data variability i.e. a relationship to $NH_4^+$ must be present, but our results imply that it is not a determining feature in oligotrophic systems where multiple resources are limiting.

Actions for revision
- We will include a condensed version of this discussion point in a revision.

**We respond to specific comments below.**
RC1: BG-2021-184

Comments on "Nitrification in the oligotrophic Atlantic Ocean"
General comments:
This study presents a high horizontal resolution dataset of ammonia oxidation (AO) rate over a 13, 000 km transect of the Atlantic Ocean. The dataset could be valuable to improve our understanding of the interaction between AO and environmental factors in the open ocean, and to facilitate the oriented-development of ocean biogeochemical models. However, the methodology, structure of this manuscript, the presentation of the results, and the Discussion need to be further condensed and reorganized. The comments are as follows:
Response
- The overall length of the manuscript is within the guidance limits for Biogeosciences.
- This comment is confusing; *'However, the methodology, structure……condensed and reorganized.'* We anticipate that the correct structure for the sentence would be; *'However, the methodology, structure of this manuscript, the presentation of the results [insert specific guidance], and the Discussion need[s] to be further condensed and reorganized.'*

Actions for revision
- We address specific suggestions as outlined below.

Specific comments:
[1] 1)By measuring the NO2–concentration and nitrogen isotope before and after the incubation, the dilution method of obtaining the ammonia oxidation rate contains some issues worthwhile discussing : there are some potential influencing factors for the NO2–product that are not carefully discussed in the manuscript. For example, the incomplete assimilatory reduction of nitrate by phytoplankton can produce NO2- especially in light condition, probably resulting in AO overestimation in this study. [2] On the other hand, NO2- consumption term during the incubation can be derived, e.g., oxidation by nitrite oxidizing microorganisms and assimilation by phytoplankton. [3] (2) The equations for calculating the AO rate were not provided in the manuscript.
Response
[1]Having published the full details of the isotope dilution method (Marine Chemistry, 2006, 2007), we provided an overview of the essential information in this MS. We summarised the issues raised by the reviewer in line 361; *'…the isotope dilution approach adopted here does not discriminate between the processes contributing to $NO_2^-$ regeneration (ammonium oxidation, algal $NO_2^-$ release and potentially a photolytic route which produces $NO_2^-$ from $NO_3^-$ (Zafiriou and True, 1979) and humic substances (Kieber and Seaton, 1999)).'* Consequently, we acknowledge this limitation of the method. We then go on to consider the potential contribution to $NO_2^-$ regeneration from the release of $NO_2^-$ by phytoplankton; *'The release of $NO_2^-$ by phytoplankton is most frequently associated with an intracellular resource imbalance linked to dynamic transitions (e.g. vertical mixing across a light gradient; Santoro et al., 2013; Lomas and Lipschultz, 2006; Meeder et al., 2012) which indicates that ammonium oxidation likely dominated the contribution to $R_{NO2}$.'*. This statement, supported by robust studies, indicates that under the experimental conditions we use (note that there are no light field transitions in incubation bottles) it is a highly reasonable expectation that $R_{NO2}$ measurements are dominated by ammonium oxidation. We expand upon this

final point with additional text to be included (below). However, the key point as summarised in the MS text (line 364) is that $NO_2^-$ release by phytoplankton reflects a resource imbalance brought about by dynamic transitions within a $NO_3^-$ sufficient environment (the $NO_2^-$ to be released has to come from somewhere). This is not the case for the pseudo-steady state oligotrophic open ocean. Consequently, we argue that $NO_2^-$ release by phytoplankton is a minor, and most likely, insignificant contributor to $NO_2^-$ turnover in the photic zone.

[2] We address this comment in our opening remarks **[OR-3]**. We summarise key points below;

- An advantage of the method applied is that the $NO_2^-$ concentration is directly measured before and after the incubation. It is not derived through calculation.
- The regeneration of $NO_2^-$ was approximately balanced by $NO_2^-$ consumption and was not depleted during incubations.
- We apply short incubations to minimise the influence of bottle effects.

[3] Full method details have been published Clark et al 2007.

Actions for revision

- The following text was prepared for a previous submission. It outlines our rationale for stating that $R_{NO2}$ is dominated by $NH_4^+$ oxidation and that $NO_2^-$ release by phytoplankton is a minor, and likely insignificant source. We will add a compressed and correctly formatted version of this text to the main body of the MS.

[revised manuscript text omitted]

concentrations persist at this depth (an average of 8.2±7.1 µmol L$^{-1}$ was measured) implied that it is not an important N-source for photosynthetic cells.

The culmination of this evidence across sampling depths indicates that while NH$_4^+$ oxidation likely dominated R$_{NO2}$ at 14% and 0.1% sPAR, NO$_2^-$ release by phytoplankton may have contributed to R$_{NO2}$ at 1% sPAR, although direct observations place this contribution at <7%.

- We will insert NO$_2^-$ concentration data and associated change during incubation.
- We will include the (well established and routinely applied) equations.

2. Pity that ammonium concentration, which is one of the key factors controlling ammonia oxidation rate in open ocean, was not measured to achieve their goal of "clarifying the relationship between the AO rate and environmental factors".

Response

We have addressed the NH$_4^+$ concentration data in opening remarks **[OR-4].** We add that the relationship to NH$_4^+$ concentration is not always evident as noted in the citations (lines 341, including Clark et al 2014). Our study indicates that additional factors modulate R$_{NO2}$ in the resource limited gyres and that NH$_4^+$ concentration may not be the dominant factor in this relationship.

Actions for revision

- We will underscore this argument in a revision.

3. The water-column-integrated AO rate that the authors used to explore the difference of AO rate between northern and southern hemispheres, was only three-water-layer based. The vertical resolution of this data was not enough to address the scientific questions.

Response

We did not set any question that required integrated rates of nitrification; as stated in the introduction (line 76), we aimed to understand links between ammonium oxidation and environmental factors. We achieved this through statistical analysis of simultaneously measured parameters. The emergent hemispherical distinction, as evidenced in Fig 7d, used volumetric rate data only. We used integrated rate data in Fig 8, to enable a tangible demonstration of what the hemispherical distinction meant. However, to address this point directly (raised by both reviewers), we agree that this resolution is low; 6 depths are most frequently used (Poulton et al 2006; Shiozaki et al 2016). We were unable to increase the depth resolution due to logistical constraints; mainly the balance between time demands compared to resources and our objective to achieve a relatively high resolution data set over an extended horizontal scale. We note that the data of Shiozaki et al (2016) for the oligotrophic Pacific, which used 6 depths, has an average ammonium oxidation rate of 0.5 mmol m$^{-2}$ d$^{-1}$. This value is comparable to our value for the oligotrophic Atlantic of 0.1 mmol m$^{-2}$ d$^{-1}$. Our data set enables this comparison, despite its coarse vertical resolution. Our data set is presently without comparison, > 12 years after the cruise was undertaken. It sets a benchmark, to be scrutinised by the scientific community with further investigations.

Actions for revision

- We will underscore the low resolution of profiles as a limitation.
- We will draw the comparison against the Shiozaki et al 2016 data as a benchmark for the scientific community to examine and test.

4. The literature survey was insufficient, some highly relevant literatures were not included, resulting in misleading statements. e.g. Line ~295: "there is relatively little information for the temperature sensitivity of nitrification in the open ocean", references like Baer 2014; Horak et al., 2017; Zheng et al., 2020 were recommended;

Response

We strongly disagree that there are 'misleading' statements in the MS. There is relatively little information for the temperature sensitivity of nitrification in the open ocean. The 'Web of Science' search results below offer very clear context to this statement;

Search 'Web of science' – 1970 onwards, Science citation index expanded, all fields search for the number of papers that include a specific search term;

  o   124, 612 - Primary production (marine, terrestrial).

- o 26, 724 – nitrification (includes terrestrial, riverine, marine, biotech, environmental engineering).
- o 4, 559 – nitrification and temperature
- o 2, 633 – marine and nitrification
- o 386 – marine and nitrification and temperature
- o 18 – marine and nitrification and temperature and oligotrophic
- o 0 - marine and nitrification and temperature and pelagic and oligotrophic

The search [marine nitrification and temperature and oligotrophic] returning 18 papers was dominated by microbiology hits with only 3 oceanography hits. Of these, 2 were <1999. One was relevant – Horak 2017, which states, '…a temperature effect on ammonia oxidation was observed at only two of four stations.' Further, the coastal study of Horak et al (2013) states *'Temperature did not have a significant effect on ammonia oxidation rates for incubation temperatures ranging from 8 to 20 °C,'*. Consequently, we stand by our statements as presented in the MS. We have limited our citations to the most relevant (and highly cited) publications and do not intend to add extensively to this list due to space constraints.

[Figure]

==Actions for revision==
- We will include the Horak et al 2017 reference.

Line ~315: "Evidence demonstrates that light structures the distribution of nitrifying organisms, whose activity may be suppressed within the photic zone (Shiozaki et al., 2016; Olson et al., 1981)", Xu et al. (2019) gave a thorough discussion on light effect on ammonium oxidation";
==Response==
It is unclear what the reviewer believes is misleading about this statement, which is supported by highly appropriate references.

==Actions for revision==
- None.

Line ~410: When the paper talks about the vertical distribution of ammonia oxidation, the work by Wan et al. (2018) should be mentioned. Wan et al. (2018) found the nitrification was low or undetectable in the mixed layer and increased rapidly downward around the nitracline. And the depth of the nitracline represents a robust spatial boundary between ammonium assimilators and oxidizers in the stratified ocean.
==Response==
The profile described by the reviewer is as stated in the MS (line 407) and consistent with previous observations (as extensively cited in our MS, Line 409, 410). What is misleading about the information presented? The reference recommended by the reviewer (Wan et al 2018, Ambient nitrate switches the ammonium consumption pathway in the euphotic ocean. Nature Communications) has 25 citations in 5 years; a relatively low response, especially for this journal. RC1's vertical profile description, as set out in Wan et al 2018, was noted in some of the earliest work on the subject (e.g. Ward et al, 1982) and is therefore not novel. It is unclear what the Wan et al (2018) citation adds to what has already been stated in our MS.

Actions for revision

- None.

Technical corrections:

1. Line 9: The product of ammonia oxidation is nitrite, not nitrate.

Response

Obviously. It is the basis of the method we use. The line reads; '*Ammonium oxidation, the first stage of the nitrification process, directs ammonium derived from organic matter decomposition towards the regeneration [of] nitrate, an important resource for photosynthetic primary producers.*' i.e. Ammonium oxidation is the first step of a sequential process that results in nitrate regeneration. We do not state that nitrate is the product of ammonium oxidation. Our text is factually correct.

Actions for revision

- We note that the word 'of' is missing from the sentence (line 9) and will address this in a revision.

2. Line 58-60: "Evidence identifies roles for light, NH4+ concentration, organic matter flux and competitive interactions with phytoplankton (Fawcett et al., 2015; Shiozaki et al., 2016; Olson, 1981; Smith et al., 2014; 2016; Bresseler and Boyd, 2009; Newell et al., 2011)." is suggested to change it to "Evidence identifies roles for light (Olson, 1981; Shiozaki et al., 2016; Xu et al., 2019), NH4 + concentration (Smith et al., 2016; Xu et al., 2019), organic matter flux (Ward and Zafiriou, 1988; Newell et al., 2011) and competitive interactions with phytoplankton (Fawcett et al., 2015; Smith et al., 2014)."

Actions for revision

- We will re-construct this sentence to indicate which references are associated with specific items of information.

Moreover, the paper Buesseler and Boyd (2009) focused on processes that control particle export and flux attenuation in the open ocean, not relevant to nitrification.

Response

We see the point being made by the reviewer – the Buessler and Boyd citation is a model development of factors influencing flux rates. Our argument was that particle flux is a modulating factor for the vertical distribution of N-regeneration. A more appropriate citation would be Benner and Amon, 2015.

Actions for revision

- We will use a more appropriate citation for this point.

3. Line 190-199: To make a clear description of results, please match the figure to the corresponding text, e.g. "The dissolved oxygen concentration ,,,, where concentrations approaching 100 μmol L-1 were measured (Fig. 3d)".

Actions for revision

- Text can be inserted to refer to specific panels.

4. In Figure 3a-d, squares stand for samples from 1% sPAR depths, while in other figures (2 and e-f) squares represent 0.1% sPAR.

Response

We thank the reviewer for highlighting this inconsistency.

Actions for revision

- We will swap Fig 3 (e, f) and Fig 2 triangle/square to be consistent with panels (a-d) of Fig 3.

5. Line 214: show the profiles of cyanobacteria Synechococcus and Prochlorococus or delete ", in contrast to the profiles of the photosynthetic cyanobacteria Synechococcus and Prochlorococus (results not shown)".

Actions for revision

- We will include these profiles in the Supplementary information. They did not co-vary with any of our measurements and do not feature in the statistical analysis of results.

6. Keep the format of all the secondary headings consistent. For example, "3.3. Microbial cell distribution and abundance." ended with a period, while secondary headings of "3.4., 4.2.-4.4." did not have periods.

Actions for revision

- We will remove all sub-heading periods to make them consistent.

7. [1] Line 219: Figure 6 did not show the vertical profile of ammonium oxidation rate; [2] The rate unit in the text is nmol L-1d-1, different from nmol L-1h-1 in the Figure 7. [3] There were two Figure 7f in the figure 7 (Line 830).

Response

[1] This was a PDF document generation error, corrected and posted to the BGS discussion page. Unfortunately, the reviewer had not noticed the posting of our corrected document.

Actions for revision

- [1] This has been corrected.
- [2] We thank the reviewer for noticing this discrepancy. Data in Fig 7(a-d) are nmol. $L^{-1}$. $d^{-1}$. The units are incorrect (nmol. $L^{-1}$. $h^{-1}$). We will correct this in a revision.
- [3] We thank the reviewer for noticing this discrepancy. Fig 7 panel (e) will be identified.

8. [1] Line 223-224: What's the meaning of "the concentration of ammonium-N oxidized"; [2] Provide the full name of "NAG" and "SAG".

Response

This section calculates an integrated value for the amount of nitrogen (as ammonium-N[itrogen]) that is oxidised to $NO_2^-$ in the same way that the unit 'nmol-N $L^{-1}$ $d^{-1}$' refers specifically to the nitrogen. [1] Our view is that the meaning of this statement is clear.

Actions for revision

- [2] NAG/SAG – We will define the North/South Atlantic Gyre.

9. Line 234 and Figure 9 would be misleading when describe the PC2 as "representing latitudinal variation", PCn just representing the combinations of the linear transformations of variables without explicit meaning. Furthermore, it is hard to infer the implications of PC1 and PC2 from 2-way ANOSIM results.

Response

The combinations of linear transformations of the variables may not have explicit meaning, but it would make no sense to do the analysis if the results are not to be interpreted. While neither sPAR bands nor latitude play any role in the placing of samples on the ordination plane, it is clear that the first PC is separating samples predominantly on the basis of sPAR bands, not latitude. Latitudinal variation is clearly playing a role in sample separation along PC2, but not PC1. PC1 is constrained to explain more variance than PC2. This means that depth variation is a more important factor than latitude in determining differences. The ANOSIM tests confirm that depth is a more important factor than latitude in determining differences among samples.

Actions for revision

- Replace:
  'PC1, separating the sPAR bands, explained 42.7% of the total variation while PC2, representing latitudinal variation, explained 23.4%.'
- With:
  'Variation on PC1, accounting for 42.7% of the total, is clearly associated with differences among sPAR bands. Differences among samples along the transect within sPAR bands are clearly associated with variation on PC2, accounting for 23.4% of the total.'
- Modify legend of figure to read:

Figure 9. Principle component analysis (PCA) ordination, highlighting the clustering of samples from specific light equivalent depths. Intersample variation on PC1 is associated with differences among the sPAR bands, and explained 42.7% of the total. Intersample differences along PC2 reflect latitudinal variation along the transect, and explained 23.4%. In combination, ~65% of variability was explained, leaving ~35% unexplained.

- Now published in full:

Somerfield PJ, Clarke KR, Gorley RN (2021) Analysis of Similarities (ANOSIM) for 2-way layouts using a generalised ANOSIM statistic, with comparative notes on Permutational Multivariate Analysis of Variance (PERMANOVA). Austral Ecology 46: 911–926 doi:10.1111/aec.13059

10. [1] The result 3.6 and Figure 10 can be moved to supplementary information. [2] And the title of color bar of Figure 10 (a) and (c) should be add on the figure.

Response

[1] We disagree. The prominence of the rates associated with these stations (28/39S) needed to be explained. They represent the first clear links between mesoscale eddy features and nitrification impacts which we believe justifies their inclusion in the main text.

Actions for revision

- [1] We do not propose to move these items to SI.
- [2] Will address this point in a revision.

11. Line 264-265: The Hovmöller diagrams should be shown to prove the conclusion that "these features had travelled from the Agulhas Retroflection, taking 2.5 years to get to this location."

Actions for revision

- We will put a Hovmöller diagram in the SI.

12. Line 275-285: Delete the paragraph of 4.1 "Overview and objectives" or move to introduction.

Response

This is a subjective comment – the paragraph provides an overview to the following discussion.

Actions for revision

- We do not propose to change this text.

13. Line 315-320: This paragraph is more than summarized the light effect on nitrifier without linking to their own data.

Response

The meaning of this sentence is not clear. The chemical/physical/biological context sections (lines 286- 395) introduce the factors that are considered in the following analysis, drawing links from the literature to previously demonstrated impacts upon nitrification.

Actions for revision

- None.

14. Tremendous typos and mistakes, e.g., Line 323: "Fig. 3.a, b, f" should be "Fig. 3a, b, f"; Line 364: "(Kieber and Seaton, 1999))" should be "(Kieber and Seaton, 1999)"; Line 372: "Fig. 4d" should be "Fig. 4b"; Line 373: "(Ward, 2002, Beman et al., 2008; 2012)" should be "(Ward, 2002; Beman et al., 2008, 2012)"; Line 447: "Shiozaki et al., (2016)" should be "Shiozaki et al. (2016)"; Line 484: "(Fig. 4d; 7a-c), " should be "(Fig. 4d, 7a-c)"; Line 493: "Nencioli et al., (2018)" should be "Nencioli et al. (2018)"…

Response

This really is an overstatement. Line 364 (Kieber and Seaton 1999) is correct as stated in the MS. A double closure is required as the reference appears within a bracketed statement '…*(ammonium oxidation, algal NO2- release and potentially a photolytic route which produces NO2- from NO3- (Zafiriou and True, 1979) and humic substances (Kieber and Seaton, 1999))…*' The remaining 'Tremendous typos and mistakes' refer to the occasional incorrectly inserted period or semi-colon.

- We will correct this issue.

15. Line 409: "a decline in rates was measured as depth increased below 1 % sPAR (Fig.6; 7d, e;", there is no rates data in Fig. 6.

As stated this was a PDF conversion issue and was corrected on the BGS discussions web-site.

- None.

16. Line 509-510: "Considering all rate data, results implied that more active nitrogen cycling took place in the South compared to the North Atlantic Gyre." This conclusion may not true since the NH4 + concentrations in South and North Atlantic Gyre may be different.

This is a confusing statement. Our MS statement was based on the integrated transect results and demonstrated a clear distinction between hemispheres. Perhaps RC1 is implying that our statement is too broad? i.e. that we measure a specific process and not N-cycling in its entirety? We can be more specific in this statement, but the implication is relevant to the wider N-cycle as the process does not take place in isolation. i.e. N-cycle processes must be approximately balanced over appropriate scales of space/time, otherwise, intermediates would accumulate. In the aphotic zone where $NO_3^-$ assimilation is zero, $NO_3^-$ accumulates. This is not so for the photic zone, pointing to a broader distinction in the way that the N-cycle operates between gyres. This is the point we were attempting to make.

- We will improve the clarity of the point we wish to make.

17. Nearly the same sentences occur repeatedly, e.g., Line 221-224 vs Line 417-419; description about labile dissolved organic matter (Line 453-455 vs 520-524)…

- ?

- None.
* * *
**We respond to specific comments below.**
**RC2: BG-2021-184**

Review #2
Reviewer report
Peer review report on "Nitrification in the oligotrophic Atlantic Ocean"
1. General comments
The manuscript 'Nitrification in the oligotrophic Atlantic Ocean' by Clark et al. investigated the NO2- production rate in the euphotic zone across a large area of the Atlantic ocean. Additional environmental variables were presented to examine the potential controls on this key process in the sunlit ocean. They found NO2- production was active at all depths, and across the study areas with large spatial variability, a striking feature of rate difference between the two hemispheres was highlighted. The authors identified that Chl-a, duration of the light phase, and Si concentration best explained the observed rates. While such large-scale observation on the biogeochemical rate is encouraging and

inspiring, there are several key issues that need to be further clarified and resolved before drawing the conclusions presented in the manuscript:

1.1 Methodology
Instead of using the 15NH4+ labeling and dark (or paired light-dark) incubation, the authors chose the 15NO2- dilution method, and the incubation was carried out under light conditions, this inevitably induces several issues:

==Response==
'Instead' implies that there is an accepted strategy for the measurement of these rates. There are in fact many ways that stable isotope approaches can be used to measure this single process. All of them have advantages and disadvantages. We have used both shorter 9 hour (as presented here) and longer 24 hour (Clark et al 2008) dilution incubations in the oligotrophic Atlantic Ocean. The average hourly rate is comparable between these two studies, as highlighted above.

==Actions for revision==
- We will include this comparison in a revision.

1) [1] The measured rate stands for the total NO2- production rate rather than ammonia oxidation rate. [2] Specifically, in the sunlit ocean, it would be expected that phytoplankton also play a significant role in producing NO2-, particularly in the upper euphotic zone where NO2- production might be dominated by phytoplankton rather than nitrifier. For this reason, I would suggest re-consider the measured rate as 'NO2- production rate' rather than 'nitrification rate.'

==Response==
[1]As the reviewer points out, it is technically correct that the method measures the rate of $NO_2^-$ production. [2] '…it would be *expected* that phytoplankton….upper euphotic zone… ' So where is the evidence for this?  Where is the $NO_3^-$ coming from to support $NO_2^-$ release? In terms of the physiology, there are critical conditions that must be met for photoautotrophic cells to release $NO_2^-$ (Lomas et al 2000). There must be sufficient, if not excess $NO_3^-$ to support N-assimilation ($NO_3^-$ is typically < 5 nmol.L$^{-1}$ in the photic zone of the oligotrophic Atlantic and is essentially limiting). Having transported $NO_3^-$ into the cell and reduced it to $NO_2^-$, a resource imbalance must be induced, otherwise $NO_2^-$ is further reduced to $NH_4^+$ and incorporated into DON (amino acids). The resource imbalance is linked to a change in environmental conditions, such as vertical mixing (i.e. a decrease in light and hence reducing power to complete the reduction of $NO_2^-$, resulting in the release of $NO_2^-$ from the cell – $NO_2^-$ is toxic and cannot be stored intracellularly, unlike $NO_3^-$). Conversely, under conditions of excess light and $NO_3^-$ sufficiency, $NO_3^-$ reduction to $NO_2^-$ and its subsequent release can be used as a mechanism to dissipate excess photosynthetic excitation energy (avoiding the production of superoxide radicles and associated cellular damage). Neither of these mechanisms will operate in the pseudo-steady state of the euphotic and oligotrophic Atlantic Ocean. The process, environmental conditions and implications of $NO_2^-$release are eloquently demonstrated in the study of Qutob et al 2002, examining seasonal transition in water column structure;

*Phytoplankton drives nitrite dynamics in the Gulf of Aqaba, Red Sea. Mutaz Al-Qutob, Clivia Häse, Max M. Tilzer and Boaz Lazar Marine Ecology Progress Series, 239, 233-239.*

> *This study focuses on the seasonal changes in the Gulf of Aqaba, Red Sea, in nitrite concentration and their relationship with phytoplankton activity, which is mainly controlled by an alternation of water-column stratification with vertical mixing. Within the euphotic zone, during thermal summer stratification, nutrient depletion was severe, and no nitrite could be detected in the upper 70 m. However, during stratification, nitrite was always associated with the nutriclines and formed a deep maximum at the bottom of the euphotic zone. In contrast, nitrite accumulated in the mixed water column during winter, closely paralleling the development of phytoplankton biomass. In the Gulf of Aqaba, maximum nitrite accumulation occurred when winter mixing reached its greatest depth, which in turn was coincident with the height of the phytoplankton spring bloom. Thus, our field data suggest that accumulation of nitrite is associated with nutrient-stimulated phytoplankton growth. This hypothesis was supported by nutrient-enrichment bioassays performed concomitantly: only when phytoplankton growth was stimulated by nutrient additions, did nitrite accumulate in the water. In the bioassays, the time-course of nitrite accumulation closely paralleled the development of phytoplankton biomass during the incubation period. We therefore suggest that the accumulation of nitrite in the mixed water column during winter is due to excretion by algal cells. Our field and experimental data show that between 10 and 15% of the total amount of nitrogen entering the mixed-water column is released as nitrite by phytoplankton.*

To summarise, the photic zone of the open Atlantic Ocean is extremely nutrient poor, it is stable and supports extremely low biomass of photosynthetic cells. The conditions for $NO_2^-$ release are not supported. Deeper in the water column at the base of the photic zone, it is more likely that $NO_2^-$ release makes a contribution to $R_{NO2}$ (there is sufficient $NO_3^-$ combined with diel transitions in the light field). However, as we reason above (text added to MS), low light supresses N-assimilation rates, such that $NO_2^-$ release will also be extremely low.

Actions for revision
- [1] We will retitle the MS as 'Presumptive nitrification in the …' Throughout the text we define and refer to $R_{NO2}$ – the rate of $NO_2^-$ regeneration. We will make it clear that this is what the method measures and that it is our reasoned assumption that $R_{NO2}$ is dominated by $NH_4^+$ oxidation.
- [2]As stated above, we will add text, which outlines our rationale that $R_{NO2}$ is dominated by $NH_4^+$ oxidation and that $NO_2^-$ release by phytoplankton is a minor, and likely insignificant source.

2) The lack of rates been measured in dark conditions would also cause a significant bias of NO2- production rates in the sunlit ocean as both phytoplankton and nitrifiers are sensitive to light.

Response
We have addressed this issue **[OR-2], [OR-3],** with key points repeated below;
- All (stable isotope) methods used to derive biological rates have advantages and weaknesses. They all introduce bias. This needs to be acknowledged (which we do).
- Both short (9h) and long term (24h) incubations applying this method returns comparable rates.
- There is no evidence of $NO_2^-$ depletion during incubations – $NO_2^-$ sources remain in balance with $NO_2^-$ sinks at all depths which span a range of light levels. i.e. the potential for $NO_2^-$assimilation by phytoplankton in the photic zone is not apparent*.

Should there be any inhibition of nitrification in the photic zone, the resultant underestimation of real rates in this study would only underscore the significance of this process in the open ocean.

Actions for revision
- We will add text (as outlined above) to present our rationale for $R_{NO2}$ being dominated by $NH_4^+$ oxidation.
- We will incorporate arguments presented (*) into the MS.

3) I am also concern about the potential effect of NO2- consumption by phytoplankton in the nutrient-deprived surface ocean on the rate measurement and calculation.
Response
This point has already been addressed immediately above and elsewhere **[OR-3]**.

Actions for revision
- We will present evidence that $NO_2^-$ is not depleted at any sampling depth during incubations.

1.2 Result and discussion
1) The lack of in-situ NH4+ measurement is another major flaw of the current study. The substrate concentration is one of the most fundamental controls for ammonia oxidation in the marine environment, so it would be of great relevance in understanding the environmental control on this process. Is that possible for the authors to

Response
To re-iterate, the term 'major flaw' is not used anywhere other than here. We have addressed the issue of $NH_4^+$ concentration data in opening remarks **[OR-4]**. The comment was unfinished by the reviewer.

Actions for revision

- None.

2) The main finding—a significant hemispheric difference of the NO2- production rate is exciting. More detailed information and discussion are encouraged further to explore the potential mechanism of this novel finding. Either it is attributed to spatial or seasonal variation, the production, and remineralization of organic matters might be a fundamental control. Thus, I suggest expanding the discussion on this point to examine this hypothesis, i.e., to examine the productivity (satellite, Bio-Argo, literature…), the strength of remineralization, etc.
Response

The hemispheric distinction was an emergent insight and not an issue that we originally intended to address. However, we thank the reviewer for acknowledging its significance. We have considered many ways of expanding upon this finding in previous versions of the manuscript, only to be criticised for speculating beyond the data. The limitation is that we have observational data from only one transect that makes this trend apparent, and limited supporting data of relevance.

In terms of the potential mechanism, our statistical analysis pointed to an aspect of organic matter production and decomposition, although there is no apparent trend in integrated primary production that reflects this observation, (data archived at BODC). We strongly suspect that the influence is more subtle, and not detectable at the level of remotely sensed chlorophyll (for example, otherwise it would surely have been reported before). However, prompted by this reviewer's enthusiasm for our observation, we contacted colleagues to discuss this aspect further. We discussed the work of Robert Letscher (Letscher et al 2013, 2016) with Dennis Hansell (University of Miami). From this discussion, we note the following;

- As stated in the MS, $NH_4^+$ oxidation requires a source of $NH_4^+$, which is most readily provided by the decomposition of labile/semi-labile Dissolved Organic Matter, formed and released by photosynthetic cells. Results of Letscher et al (2013) indicate that there is an east-west gradient in [DON] whereby DON produced in the EBUE of the north and south Atlantic is consumed within the photic zone during westwardly lateral transport. Upwelled water on the eastern Atlantic flanks is vertically mixed and/or subducted during winter mixing to the deep euphotic zone, below the mixed layer. Over the subsequent months to years, DON is laterally transported within the deep euphotic zone to the gyre interior.
- Therefore,
  - The remineralisation of EBUE DON represents a new source of N to the Atlantic photic zone. $NH_4^+$ regeneration and nitrification make this DIN available to primary producers, which is quantitatively similar in magnitude to vertically mixed DIN (Letscher et al 2016).
  - The strength and productivity of upwelling is greater in the Benguela than the Mauritanian upwelling (primary production data), leading to a measurable difference in the down-stream [DON] (Letscher et al 2013). By speculative extension, we hypothesize that it is this difference, in combination with lateral transport, that supports the interhemispheric difference in AO rate.
  - Seasonal variability in the strength of upwelling and winter mixing will influence the distribution and rate of N-remineralisation activity across extended scales within Atlantic gyres.
  - The activity reflects the recovery of limiting resources (N and P) via microbial transformation of DOM; i.e. the MCP, in the subtropical oceans which are regions of large scale downwelling (and hence C-export).
  - It is implicit that in addition to supporting nitrification, there will be the associated production of $N_2O$ that will be greater in the SAG than the NAG (whether or not this is released to the atmosphere).

Actions for revision
- We will review our comments on this aspect with a view to expansion, while ensuring that we present our reasoned speculation coherently.

3) The authors declared that NO2- production was enhanced in the mesoscale eddies. [1] While I appreciate that mesoscale processes exert profound influences on marine biogeochemistry, more evidence is expected to bolster this argument. [2] For instance, mesoscale eddy is a ubiquitous feature in the global ocean, have the authors examined

whether they sampled in other eddies aside from these two eddies reported here? [3] Can the authors expand a bit to examine the underlying causes? i.e., Increased primary production? Intensified remineralization? Alleviated competition between nitrifier and phytoplankton etc.

Response

[1], [3] The eddy stations represent 2 profiles of 33 stations. Only one of the eddy features (28S) was remotely generated and traversed the SAG to influence its biogeochemistry across an extended scale.  As we state, one profile at the edge of a 300km diameter feature is in no way representative – results may only represent the eddy fringing region. Any kind of apparently sustained (given the age of the feature) stimulation of productivity due to eddy induced mixing will ultimately lead to increased DOM production and remineralisation. We state this in lines 498-503. However, we do not have information about competitive interactions or rates of primary productivity to inform this argument any further. Although of interest, not least because of the link to $N_2O$, groundless speculation will weaken the MS. We attempted this with previous submissions (PNAS/GRL) and were roundly criticised for it. This version of the MS stays well within the data; a strategy we wish to retain.

[2] We state (line 489) that  the techniques used to identify eddy features were used to examine all stations along the transect. None of the other stations were associated with eddy features at the point of sampling.

Actions for revision

- We will review our comments on this aspect with a view to limited expansion, while ensuring that we present our reasoned speculation coherently.

4) I am also curious about the effect of light, i.e., any explanation for the lack of correlation between the measured rate vs. light intensity, but the duration of light can well explain the observed rates?

Response

The statistical analysis demonstrated that the greatest fraction of co-variability in $R_{NO2}$ could be explained with significance through the combination of chlorophyll, light duration and silicate. This was not to the exclusion of light intensity; but rather that combinations that included light intensity were less significant.

Actions for revision

- None

Also, it is counterintuitive to see the positive correlation between light duration and NO2- production rate.

Response

Perhaps we were not completely clear in our discussion of this aspect. Correlation does not imply causality. We are not stating that there is a direct link between light duration and $R_{NO2.}$ We are stating that they co-vary. We reason that they are linked through an intermediate - a product of photosynthetic activity; i.e.  DOM, which is released by photosynthetically active cells. DOM release is a ubiquitous process, with 2-50% of photosynthetically fixed carbon released as DOM either passively or actively (Thornton 2014). In both lab and field studies, elevated quantities of DOM (i.e. in excess of basal rates) are released within hours of light exposure (Mühlenbruch et al, 2018), underscoring the direct link between photosynthetic cellular activity and labile DOM availability. Labile DOM is utilised by bacteria within minutes to hours (Jiao et al 2010). The rapid decomposition of labile DOM regenerates $NH_4^+$ (and $CO_2$, $PO_4^{3-}$), the 'substrate' for $NH_4^+$ oxidation. This illustrates the mechanism we propose. Finally, we argue that silicate is a tracer for short term mixing events across the nitricline, initiating and supporting the sporadic (i.e. short-term) growth of primary producers. Such events disrupt the system and imply that nitrification may respond to short-term physical instability that erodes biogeochemical gradients, making it difficult to directly link observations to extant conditions (as activity reflects the very recent past, rather than the present).

- We will review our comments on this aspect and attempt to clarify the arguments, using the points raised by the reviewer and our response.

**2. Specific comments**
Line 1: You do not need the full stop at the end of the tile.
Actions for revision
- We will address this.

Lines 26-27: Why not NH4+?
Actions for revision
- We will add the text 'which ultimately regenerate $NH_4^+$' to this line.

Lines 91-95: I encourage the authors to expand to clarify the method, particularly the detection limit and the accuracy of the trace level of nutrient measurement.
Response
The method we apply converts ambient $NO_2^-$ to a compound that is physically collected and analysed. At nanomolar DIN concentrations, higher seawater volumes are collected to satisfy detection limits, although these are always within practical limits (typically only 200mL seawater at $5nmol.L^{-1}$ is required). Measuring nmol concentrations of DIN within modest seawater volumes is well within the analytical capabilities of GCMS systems.

Actions for revision
- We will include key facts about the methods performance and limitations in a revision.

Line 106: What criteria are used to define the euphotic zone, 1% PAR or 0.1 PAR%? )
Response
By definition, the fact that light is still measured at 0.1% implies that it is within the euphotic zone. We intended to define the 0.1% sPAR as the base of the photic zone, although inconsistencies in our text were identified during the development of this MS, which should have been removed.

Actions for revision
- We will ensure that we are consistent with this definition throughout the text, stating that 0.1% sPAR represents the base of the photic zone for the purposes of this study.

Line 109: Taken the extremely low level of NO2- outside the PNM layer (i.e., less than 10nM in many samples?), is that mean that only <1nM of 15N-NO2- (<10%) was added into the incubation system? Would that cause bias if the trace amount of NO2- was assimilated by the phytoplankton and/or bring a significant challenge in measuring the NO2- isotope at such low concentration?

Response
Points addressed previously –
- Biological sinks ($NO_2^-$ oxiders or potentially $NO_2^-$ assimilators) will not selectively remove the tracer. While there is isotopic discrimination at the level of enzyme kinetics, it is only associated with resource sufficient conditions.
- Short incubations limit potential errors associated with tracer studies.
- There is no depletion of $NO_2^-$ during incubations as now evidenced by data.
- With this method, the analyst thinks in terms of the amount of N in the sample rather than the ambient DIN concentration. The volume of seawater sample is adjusted to achieve the N-content required for the analysis. At high DIN, lower volumes are required, and visa versa. Consequently, this marine system does not present a challenge to the analytical detection limit (Clark 2007). Genuinely trace $^{15}N$ additions are used and readily detected (line 118).

- None.

Line 116: What is the reason for performing a ten h day-time incubation? Taking the inhibition of ammonia oxidizers by light in the sunlit ocean and their ability to recover under dark, ten-hour day-time incubation rather than a full day incubation covering the light-dark cycle would cause underestimating ammonia oxidation rate. In another aspect, NO2-release during assimilatory NO3- reduction appears to be another key source of NO2- in the euphotic zone. The dilution method used in this study cannot distinguish these two sources of NO2- and thus would overestimate the ammonia oxidation rate.

Response

All of these comments have been raised by this reviewer previously and addressed.

Actions for revision
- None.

Lines 120-128: The method used here for NO2- isotope analysis is quite complex that involves multiple steps and reagents. Extreme care should be taken to avoid any contamination for the isotope analysis of the trace amount of NO2-. For this reason, some introductions on the detection limit and accuracy will benefit the audience on the accuracy of the result.

Response

The challenge calls for sophisticated approaches. The expectation that significant advance can be made with crude approaches has thankfully passed. We developed and applied these methods across a range of marine systems for over 15 years. The publications, which contain all the details requested by the reviewer, have been highly cited.

Actions for revision
- We will include key facts of the method in a revision of the paper.

Lines 135-139: Similarly, I suggest further clarifying the calculation process and detection limit of the rate.
Response
The calculations are well established and routinely applied for isotope-based incubations. Details are presented in our previous publications, to which we refer.

Actions for revision
- Already addressed.

Lines 175-176: Citation required?
Response
The definition used was not consistent with the more commonly used one for "Mixed Layer Depth", so we will use a different term to explain that this is a measure indicative of the overall upwelling/downwelling associated with mesoscale features.

Actions for revision
Use alternative wording that does not lead to confusion with others' definitions.

Lines 187-189: [1] Perhaps some description on the integrated light intensity (dose)? [2] And the potential reason for the absence of a corresponding increase of irradiance?

Response
[1] The meaning or relevance isn't clear. We do not present any information about integrated light intensity. We present light dose, defined in the Fig 2 legend as PAR x light duration. It provides an indication of the energy delivered to each sampling depth and is a relatively straight forward concept. [2] We do not show any data for irradiance. If the reviewer is referring to the anticipated increase in light dose moving progressively south (i.e. in parallel with increased light duration), this would make more sense. The light dose data is presented on a logarithmic scale, which makes such differences harder to discern. It is clearer at 0.1% sPAR (Fig 2c), although not at the shallower depths.

Actions for revision
- None.

Lines 213-214: Any reason for not showing the results of Synchronous and Prochlorococcus? Given that these cyanobacteria are the most representative and abundant phytoplankton in the oligotrophic ocean and the main competitors for NH4+, information on the distribution of Synchronous and Prochlorococcus should be relevant of interpreting the observed ammonia oxidation rate.

Response
The data is now included in SI.

Actions for revision
- Profile data added to SI.

Lines 220-221: Since the vertical resolution is relatively coarse, it would not be robust to do the depth-integrated rate calculation.

Response
The vertical resolution for $R_{NO2}$ is indeed coarse. We address this point above.

Actions for revision
- As set out in the first instance of this point being raised.

Lines 305-308: Although there is a clear spatial pattern of the light duration of the investigated area, the light dose did not show a similar pattern. Additional analysis between the rates and light dose might also be helpful.

Response
See response above on this topic – there is a response, but the logarithmic scale makes this harder to discern. Note that light duration only increased by 3 hours, while instantaneous sPAR measurements will be influenced by cloud cover etc. Light dose is already a factor in the statistical analysis – what justification is there for considering only light and $R_{NO2}$?

Actions for revision
- None.

Lines 318-320: Numerous studies have discussed the effect of light and the competition between nitrifiers and phytoplankton in the sunlit ocean. It deserves to discuss further the potential effect of the diel cycle and the role of phytoplankton on consuming and /or producing NO2-.

Response
This point has been raised multiple times by the reviewer, and addressed above and [OR-2], [OR-3]

Actions for revision

- Text to SI.

Lines 336-342: Regretfully, that NH4+ concentration was not measured in this study! The substrate acts as the most critical factor in regulating the ammonia oxidation rate in the ocean. The significant correlation between substrate concentration and ammonia oxidation rate is more often reported in the literature, so it would be better to point it out. Also, the NH4+ concentration is usually very low (i.e., lower than the reported Ks of the ammonia oxidizers) in the oligotrophic gyres; saturation kinetics appears to be very unlikely in the study area.

==Response==
The lack of $NH_4^+$ is a frustrating shortfall, which we have addressed elsewhere **[OR-4]**. A correlation between $NH_4^+$ concentration and oxidation is demonstrated in systems typified by higher rates of biological activity. Under oligotrophic conditions, multiple points of resource limitation likely mean that this link is considerably weaker and less significant. No such relationship was demonstrated in the study of Clark et al 2008, or in the study of Shiozaki et al 2016 in the oligotrophic Pacific. We are very aware that $NH_4^+$ concentration is extremely low in the oligotrophic ocean. While the kinetics of $NH_4^+$ oxidation are unlikely to be saturated in the study area, the point being made is that the process does not respond linearly to $NH_4^+$ concentration, as is implied by the specific rate of nitrification ($d^{-1}$).

==Actions for revision==
- None.

Line 361-362 This is another key issue of the study. A main weakness of the dilution method lies in its unavailability of discerning the potential sources. It is very likely that the rate of NO2- production by ammonia oxidizers in the euphotic zone is overestimated, particularly at the upper euphotic zone where nitrifiers are inhibited by light and outcompeted by the phytoplankton, the NO2- production rate measured here is more likely attributed to phytoplankton rather nitrifiers.

==Response==
'...more likely attributed to phytoplankton rather nitrifiers...' Evidence? We have addressed these points at multiple stages in this response, supported by robust arguments, citations and new data. Just because phytoplankton are growing, they are not necessarily releasing $NO_2^-$. It is the exception that plankton release $NO_2^-$. There must an environmental transition (light/temperature etc) and there must be sufficient $NO_3^-$.

==Actions for revision==
- As outlined in previous responses to this point.

Lines 385-386: See my comment above. I suggest showing the profiles of Synchronous and Prochlorococcus.
==Response==
See our comment above in response to this comment.

==Actions for revision==
- As outlined in previous responses to this point.

Lines 386-387: Doesn't that mean the potential of NO2- released by phytoplankton might occur throughout the sampling depth and thus should be taken into account in the discussion?

==Actions for revision==
- See previous responses to the point that has been raised multiple times.

Lines 415-430: It is interesting to see the systematic rate difference between the hemispheres. The authors have proposed several potential causes for explaining the phenomenon. Nevertheless, I would like to see more detailed

information on bolstering these hypotheses[1], i.e., does the productivity/ biomass (either from Argo or satellite) show a similar spatial pattern between the hemispheres during the cruise? Counterintuitively, the duration of light increases from the northern hemisphere to the southern hemisphere. [2] As light is an inhibitor for the nitrifiers, is there any explanation for the apparent positive correlation between ammonia oxidation rate and light duration?

Response
[1] We have addressed this point.
[2] As we have outlined above, this is co-variability, not correlation. It is not a direct effect of light on nitrifiers, we speculate that it is an effect of light on photosynthetic cells that produce labile DOM, which is subsequently decomposed to regenerate the resource needed by ammonium oxidisers.

Actions for revision
- As stated above.

Lines 434-435: What does this sentence mean?
Actions for revision
- The MS states;

    'While there is strong evidence to suggest that ammonium oxidation reflected ammonium oxidation activity rather than alternative sources of $NO_2^-$, the measurement…'

    This should refer to $R_{NO2}$;

    'While there is strong evidence to suggest that $R_{NO2}$ reflected ammonium oxidation activity rather than alternative sources of $NO_2^-$, the measurement…;'

Lines 471-472: Do the bacterial abundance indicate higher remineralization rates in the South hemisphere?

Response
Cellular abundance is no indication of cellular activity.  Nothing can be implied about remineralisation rates simply from the AFC abundance of cells.

Actions for revision
- None.

Lines 483-505: It is also quite interesting to see the significantly enhanced rate in 28°S and 39°S, which was attributed to the mesoscale process. While it deserves to expand a bit more to discuss it: 1) it is true that the prominent high rate occurred in these two eddies, worth noting that mesoscale eddy is a ubiquitous feature in the ocean, [1] are there any other eddies at the sampling stations during the cruise? If so, is there any evidence of stimulation of ammonia oxidation by the eddy? 2) [2]Is that possible to look into the biomass, productivity, heterotrophic activities, directly or indirectly, to bolster the argument that the high rate was due to the eddy?

Response
[1] We state clearly in the MS that no other stations were associated with mesoscale eddy features at the point of sampling.
[2] As stated, we have insufficient information for a robust analysis of this finding, which was completely co-incidental. We have 1 profile in an eddy of interest (28S, not 39S). This simply isn't enough.

Actions for revision
- None.

Line 795: I suggest plotting the depth of the mixed layer in Fig. 2a.

Response

The depth of the mixed layer was not considered relevant to this manuscript (at no point was it discussed or implemented in any way) and so was not included within the figures. We see no justification for adding MLD to Fig 2a.

Actions for revision
- None.

Line 805: I suggest presenting nitrite and silicate (Fig. 3e and 3f) in the same way with Fig. 3a-3d, because the vertical resolution (only three depths) is too coarse for the extrapolation.
Response
All nutrient data presented in Fig 3 are measured by autoanalyzer at multiple depths throughout the photic zone. Between 9 and 12 depths are selected. For oxygen, it is a semi-continuous profile of high-resolution discrete measurements. The three sampling depths are limited to isotope studies only. The data in Fig 3 are plotted to show specific features - line plots co-incident with isotope study depths (panel a-d) while panel (e-f) show the PNM and equatorial upwelling.

Actions for revision
- The distinction between sampling depths for inorganic nutrient analysis, oxygen analysis and isotope studies will be made in a revision.
- We note that in Fig 3 there is an inconsistency between using triangles and squares for 1 and 0.1% sPAR between plots a-d and plots e-f. This will be addressed.

Line 825: How was the NO3- profile derived, given that only three depths were sampled for nutrient analysis in each station?
Response
As outlined above, additional detail will be added to the M&M section regarding the resolution of inorganic nutrient analysis. The Profile in Fig 6 was derived from a profile of 12 points, not 3.

Line 830: Some statistical analysis (i.e., if the difference between the hemispheres is significant) would further bolster the difference between the two hemispheres in Fig. 7d.
Response
This analysis is presented in the MS, line 227.

Actions for revision
None.

Line 840: Again, the depth-integrated rate in Fig. 8 was based on the rates measured at three depths, which was insufficient for extrapolation.
We have responded to this point above.

Acknowledgments.
We will amend this section to acknowledge the contribution made by reviewers and discussions with colleges which have improved this manuscript.

---

## Referee Report (RR1)

**Reviewer report on "Presumptive nitrification in the oligotrophic Atlantic Ocean"**

The revision has addressed my previous concerns on the methodology part (sensitivity and accuracy of $R_{NO2-}$ measurement under extreme low concentration); and partly answer my curiosity about the hemispheric differences of the $R_{NO2-}$. While I am not convinced by the conclusion that $NO_2^-$ production by phytoplankton is insignificant throughout the study area and all depths.

Line 1: 'Presumptive nitrification' is not clear enough to the audience. I would suggest using 'nitrite production'.

Lines 419-430: The conclusion is made on the assumption that $NO_2^-$ release during assimilative $NO_3^-$ reduction is negligible in $NO_3^-$ depleted water. i.e., the upper mixed layer of the oligotrophic ocean. However, $NO_2^-$ production via $NO_3^-$ reduction has been measured above the nitracline. The rate is higher than $NH_4^+$ oxidation in both the California Current (Santoro et al., 2013) and the North Pacific Subtropical Gyre (Wan et al., 2021), suggesting a considerable fraction of $NO_2^-$, is contributed by phytoplankton even in the nutrient-depleted water. On the other hand, $NH_4^+$ oxidation is frequently to be found at a rate 'below the detection limit' using $^{15}NH_4^+$ labelling incubation at the surface ocean (i.e., Horak et al., 2013; Santoro et al., 2013; Shiozaki et al., 2016), demonstrating extreme low activity of marine AOO in the surface layer of the oligotrophic ocean. These results indicate that at least a certain fraction of $NO_2^-$ is contributed by phytoplankton in the mixed layer.

Lines 431-453: Accumulating evidence demonstrates that $NH_4^+$ oxidation is the main source of $NO_2^-$ at the lower euphotic zone (i.e., the primary mechanism that sustains the PNM). The contribution of $NH_4^+$ oxidation to PNM ranged from ~70% to ~90% in different studies (i.e. Buchwald and Casciotti, 2013; Chen et al., 2021; Santoro et al., 2013; Wan et al., 2021). It's better to review the literature to provide a more comprehensive statement on the contribution of $NO_3^-$ reduction to $NO_2^-$ at the PNM layer. And again, the $NO_2^-$ release during assimilative $NO_3^-$ reduction is not negligible.

Lines 454-473: I agree that at a depth of 0.1% of PAR, $NO_2^-$ production should be predominated by $NH_4^+$ oxidation as the growth of phytoplankton is limited by the dim light. However, the statement that 'The fact that such elevated NO3- concentrations persist at this

depth (an average of 8.2±7.1 µmol L-1 was measured) implied that NO3- was not an important N-source for photosynthetic cells.' is not justified. The high $NO_3^-$ concentration at the subsurface water indicates that NO3- supply rate is higher than NO3- assimilation rate due to the light limitation, it cannot tell the nutrient structure (i.e. NO3- vs. NH4+ or DON) by the phytoplankton.

Lines 486-493: Inhibition of marine nitrifiers by light has been well demonstrated in numerous studies, and the rate measured in the present study (1.2±1.9 nmol/d) appears to be lower than the rate collected from 24h incubation (2.9±2.4 nmol/d). I agree with the statement that 'Results presented here may represent a lower limit for RNO2', but not for the idea that 'the exclusion of a dark phase to the incubations used here had no significant impact on average values between studies'.

Ref.

Buchwald and Casciotti, 2013. Isotopic ratios of nitrite as tracers of the sources and age of oceanic nitrite

Chen et al., 2021. Nitrite cycle indicated by dual isotopes in the northern South China Sea

Horak et al., 2013. Ammonia oxidation kinetics and temperature sensitivity of a natural marine community dominated by Archaea

Santoro et al., 2013. Measurements of nitrite production in and around the primary nitrite maximum in the central California Current

Shiozaki et al., 2016. Nitrification and its influence on biogeochemical cycles from the equatorial Pacific to the Arctic Ocean

Wan et al., 2021. Phytoplankton-nitrifier interactions control the geographic distribution of nitrite in the upper ocean

---

## Author Response (AR2)

Line 1: 'Presumptive nitrification' is not clear enough to the audience. I would suggest using 'nitrite production'.

*Changed to "Nitrite regeneration in the oligotrophic Atlantic Ocean"*

Lines 419-430: The conclusion is made on the assumption that NO2 - release during assimilative NO3 - reduction is negligible in NO3 - depleted water. i.e., the upper mixed layer of the oligotrophic ocean. However, NO2 - production via NO3 - reduction has been measured above the nitracline. The rate is higher than NH4 + oxidation in both the California Current (Santoro et al., 2013) and the North Pacific Subtropical Gyre (Wan et al., 2021), suggesting a considerable fraction of NO2 - , is contributed by phytoplankton even in the nutrient-depleted water. On the other hand, NH4 + oxidation is frequently to be found at a rate 'below the detection limit' using 15NH4 + labelling incubation at the surface ocean (i.e., Horak et al., 2013; Santoro et al., 2013; Shiozaki et al., 2016), demonstrating extreme low activity of marine AOO in the surface layer of the oligotrophic ocean. These results indicate that at least a certain fraction of NO2 - is contributed by phytoplankton in the mixed layer.

*In our defence here the paper by Santoro et al returns only one observation of $NO_2$ release from $NO_3$ in fully oligotrophic conditions. Their observation in nutrient replete conditions was reported as equivalent to a filtered control and thus there is some doubt cast over that result. The Wan et al paper was published after the submission of the previous manuscript and we did not have foresight of this data. We have now reconsidered this section and have addressed the reviewers concerns accordingly. (**Now lines 420 - 431**)*

Lines 431-453: Accumulating evidence demonstrates that NH4 + oxidation is the main source of NO2 - at the lower euphotic zone (i.e., the primary mechanism that sustains the PNM). The contribution of NH4 + oxidation to PNM ranged from ~70% to ~90% in different studies (i.e. Buchwald and Casciotti, 2013; Chen et al., 2021; Santoro et al., 2013; Wan et al., 2021). It's better to review the literature to provide a more comprehensive statement on the contribution of NO3 - reduction to NO2 - at the PNM layer. And again, the NO2 - release during assimilative NO3 - reduction is not negligible.

*We have now addressed the contribution of ammonium oxidation to the PNM and clarified a statement on the contribution of nitrate reduction. We had not stated in this section that the phytoplankton contribution is negligible. (**Now lines 436 - 444**)*

Lines 454-473: I agree that at a depth of 0.1% of PAR, NO2 - production should be predominated by NH4 + oxidation as the growth of phytoplankton is limited by the dim light. However, the statement that 'The fact that such elevated NO3- concentrations persist at this depth (an average of 8.2±7.1 µmol L-1 was measured) implied that NO3- was not an important N-source for photosynthetic cells.' is not justified. The high NO3 - concentration at the subsurface water indicates that NO3- supply rate is higher than NO3- assimilation rate due to the light limitation, it cannot tell the nutrient structure (i.e. NO3- vs. NH4+ or DON) by the phytoplankton.

*On reflection we agree with the reviewers argument here, though do not understand the very last point " it cannot tell the nutrient structure (i.e. NO3- vs. NH4+ or DON) by the phytoplankton". This section has been edited in-line with the reviewers comments. (**Now lines 456 - 471**)*

Lines 486-493: Inhibition of marine nitrifiers by light has been well demonstrated in numerous studies, and the rate measured in the present study (1.2±1.9 nmol/d) appears to be lower than the rate collected from 24h incubation (2.9±2.4 nmol/d). I agree with the statement that 'Results presented here may represent a lower limit for RNO2', but not for the idea that 'the exclusion of a dark phase to the incubations used here had no significant impact on average values between studies'.

*This section has amended according to the reviewers comments. (**Now lines 484 - 494**)*